# Lipopolysaccharide inhalation recruits monocytes and dendritic cell subsets to the alveolar airspace

Laura Jardine [1,2,4], Sarah Wiscombe[1,2,4], Gary Reynolds[1,2], David McDonald[1], Andrew Fuller[1], Kile Green[1], Andrew Filby[1], Ian Forrest[2], Marie-Helene Ruchaud-Sparagano[1], Jonathan Scott[1], Matthew Collin[1,2], Muzlifah Haniffa [1,3,4] & A. John Simpson[1,2,4]

Mononuclear phagocytes (MPs) including monocytes, macrophages and dendritic cells (DCs) are critical innate immune effectors and initiators of the adaptive immune response. MPs are present in the alveolar airspace at steady state, however little is known about DC recruitment in acute pulmonary inflammation. Here we use lipopolysaccharide inhalation to induce acute inflammation in healthy volunteers and examine the impact on bronchoalveolar lavage fluid and blood MP repertoire. Classical monocytes and two DC subsets (DC2/3 and DC5) are expanded in bronchoalveolar lavage fluid 8 h after lipopolysaccharide inhalation. Surface phenotyping, gene expression profiling and parallel analysis of blood indicate recruited DCs are blood-derived. Recruited monocytes and DCs rapidly adopt typical airspace-resident MP gene expression profiles. Following lipopolysaccharide inhalation, alveolar macrophages strongly up-regulate cytokines for MP recruitment. Our study defines the characteristics of human DCs and monocytes recruited into bronchoalveolar space immediately following localised acute inflammatory stimulus in vivo.

---

[1] Institute of Cellular Medicine, Newcastle University, Newcastle upon Tyne NE2 4HH, UK. [2] Newcastle upon Tyne Hospitals NHS Foundation Trust, Newcastle upon Tyne NE7 7DN, UK. [3] Department of Dermatology and NIHR Newcastle Biomedical Research Centre, Newcastle Hospitals NHS Foundation Trust, Newcastle upon Tyne NE2 4LP, UK. [4] These authors contributed equally: Laura Jardine, Sarah Wiscombe, Muzlifah Haniffa, A. John Simpson. Correspondence and requests for materials should be addressed to L.J. (email: Laura.Jardine@ncl.ac.uk) or to M.H. (email: M.A.Haniffa@ncl.ac.uk)

The alveoli of the lung present a large but fragile surface area to the environment. Maintaining the integrity of the alveolar-capillary membrane is critical to effective gas exchange. Immune regulation at this interface must control infection and limit immunopathology. Mononuclear phagocytes (MPs), comprising monocytes, macrophages, and dendritic cells (DCs) have a critical role at any environmental interface as innate immune effectors equipped to shape the adaptive immune response through antigen presentation, co-stimulation, and cytokine production.

Leukocytes from the alveolar airspace can be readily isolated by bronchoscopy and bronchoalveolar lavage (BAL). BAL has been less extensively characterized than lung tissue but presents a number of advantages as a window to the lung's immune system: BAL yields a cell suspension free from contaminating blood leukocytes with minimal processing requirement thus preserving surface antigens and native activation status.

MPs in any anatomical compartment are a heterogeneous group of leukocytes. Establishing the steady-state repertoire of a compartment is crucial to understanding infiltrates seen in inflammation. Most tissues contain embryonically-derived macrophages with variable contributions from circulating monocytes depending on how available the tissue niche is[1,2]. Monocyte-derived cells are observed adopting a spectrum of macrophage or DC features depending on the tissue. While steady state monocyte-derived DCs are observed in mouse skin and gut[3–5], their existence in human tissues has not been convincingly established. DCs are broadly divided into plasmacytoid DCs (pDC) that characteristically produce IFN-α and conventional DCs (cDC) that effectively stimulate T cell proliferation[6]. Two subsets of cDCs with homology across species have been clearly identified. cDC1 expresses CD141, CLEC9A, and XCR1 in humans and is adept at cross-presenting antigen[7–11]. cDC2 expresses CD1c in humans and is important for activation of CD4 T cells, induction of regulatory T cells and activation of Th2 and Th17 responses[12,13].

A number of recent studies have capitalized on multi-parameter flow cytometry to define subsets of MPs across human lung compartments[14–18], but differences in approach have led to continued debate about whether rare DC subsets (pDC and cDC1) exist in BAL or have been inadvertently excluded during analysis. As blood is a source of tissue-recruited leukocytes, a logical approach would use blood MP definitions to classify tissue MPs. Recent insights from single cell RNA sequencing have revealed additional complexity in our understanding of blood MPs, including the presence of previously undiscovered DC subsets, and heterogeneity within existing subsets[19]. Briefly, this confirmed the presence of cDC1 (DC1) and pDC (pDC or DC6). It revealed two subdivisions within cDC2 (DC2, DC3) and identified additional DCs subsets: Axl+Siglec−6+ DCs (AS DC or DC5) and CD1c-CD141− DCs (DC4). To date, this revised classification has not been tested in non-lymphoid tissue.

As our understanding of the BAL MP repertoire in steady state develops, it becomes tangible to address the question of what happens during inflammation. In mice, inflammatory macrophages and DCs have been described in numerous infection and sterile inflammation models[20–25]. Monocytes are thought to be the source of inflammatory DCs, based on studies in CCR2 and Flt3-deficient animals and adoptive transfer of monocytes[20–24]. Distinct inflammatory macrophages and DCs have also been identified in human chronic inflammatory exudates[26]. These inflammatory DCs are proposed to arise from monocytes based on transcriptional similarity to in vitro monocyte-derived DCs[26]. Early time-points of inflammation have not been adequately explored.

Here, we use inhaled lipopolysaccharide (LPS) as an experimental inflammatory stimulus to reproducibly study the earliest events in human lung inflammation. Eight MP subsets can be identified in steady state BAL following sterile saline inhalation (SS-BAL), in line with subsets described in blood. Monocytes and two myeloid DC subsets (DC2/3 and DC5) are recruited as early as 8 h following LPS inhalation (LPS-BAL) and rapidly adopt gene expression profiles characteristic of airspace MP residence. The cytokine and chemokine profile of BAL implicates AMs as the likely instigator of blood monocyte and DC recruitment into the alveolar airspace during acute inflammation.

## Results

### Accumulation of alveolar neutrophils, monocytes, and DCs.

The steady state MP repertoire of BAL was defined by flow cytometry in healthy individuals 8 h after inhalation of isotonic saline (Fig. 1a, Supplementary Fig. 1). We used a recent description of human blood MP diversity based on single cell RNA-sequencing as a template for identifying BAL MP populations[19]. As an adaption for BAL analysis, we first used side scatter and CD45 expression to exclude CD45$^{lo}$SSC$^{mid}$ neutrophils and identify alveolar macrophages (AM) as CD45$^+$SSC$^{hi}$ cells (Supplementary Fig. 1). BAL CD45$^+$SSC$^{lo}$ cells were then comparable to peripheral blood mononuclear cells (PBMC) (Fig. 1b). Peripheral blood was sourced from healthy controls (HC) that had not received inhalation challenge. CD45$^+$SSC$^{lo}$ cells that expressed HLA-DR but not lineage markers (CD3, 19, 20, 56) were gated into CD14$^{++}$CD16$^-$, CD14$^{++}$CD16$^+$, and CD14$^+$CD16$^{++}$ fractions analogous to blood classical, intermediate and non-classical monocytes, respectively. The CD14$^-$CD16$^-$ fraction containing DCs was first probed for the rare Axl$^+$Siglec6$^+$ population (DC5) as its varied expression of CD123 and CD11c would otherwise place it within pDC and cDC gates. After DC5 exclusion, CD123$^+$CD11c$^{lo}$ cells identified pDC and further segregation of the CD11c$^+$ cells using a combination of BTLA and CD1c identified DC1 and DC2/3. While DC1 is most typically defined by its expression of CD141 or CLEC9A, immune gene expression by cells sorted from the BTLA$^{hi}$ gate expressed the expected gene profile of DC1, confirming the validity of this approach (Supplementary Fig. 1). We did detect a population of CD11c$^+$CD1c$^-$BTLA$^-$ cells within our DC gate, which are likely to correspond to DC4 described in Villani et al., but can only refer to these as CD11c$^+$CD1c$^-$ cells without further characterization.

The dominant MP subset in SS-BAL was the AM (Table 1). Comparing SS-BAL CD45$^+$SSC$^{lo}$ cells with PBMC, MPs were richer in SS-BAL (Table 1). Of MPs, the most abundant subset was a CD14$^{++}$CD16$^+$ population (Table 1; Fig. 1d). CD14$^{++}$CD16$^-$ cells, analogous to classical monocytes in blood, were comparatively rare in SS-BAL and CD14$^+$CD16$^{++}$ cells, analogous to non-classical blood monocytes, were virtually absent. All DC subsets, especially CD1c-expressing DC1 and DC2, were enriched in SS-BAL relative to blood.

Following LPS inhalation, the greatest leukocyte expansion was in CD14$^{++}$CD16$^-$ MPs (400-fold difference in mean concentration between SS-BAL and LPS-BAL), followed by neutrophils and DCs (Fig. 1c). Amongst DCs, the concentrations of CD1c$^+$ DCs (DC2/3) and DC5 were selectively increased (Fig. 1c).

To examine the impact of acute lung inflammation on peripheral blood MP populations, their concentrations were tracked at 2-h intervals following inhalation of LPS or saline (Fig. 1e). Neutrophils, which were abundant in LPS-BAL and pDCs, which were not enriched in LPS-BAL, were tracked for comparison with monocytes and DCs. Following saline inhalation, no differences in leukocyte concentrations occurred compared with baseline. Following LPS inhalation, blood neutrophil concentration rose progressively, but pDC and

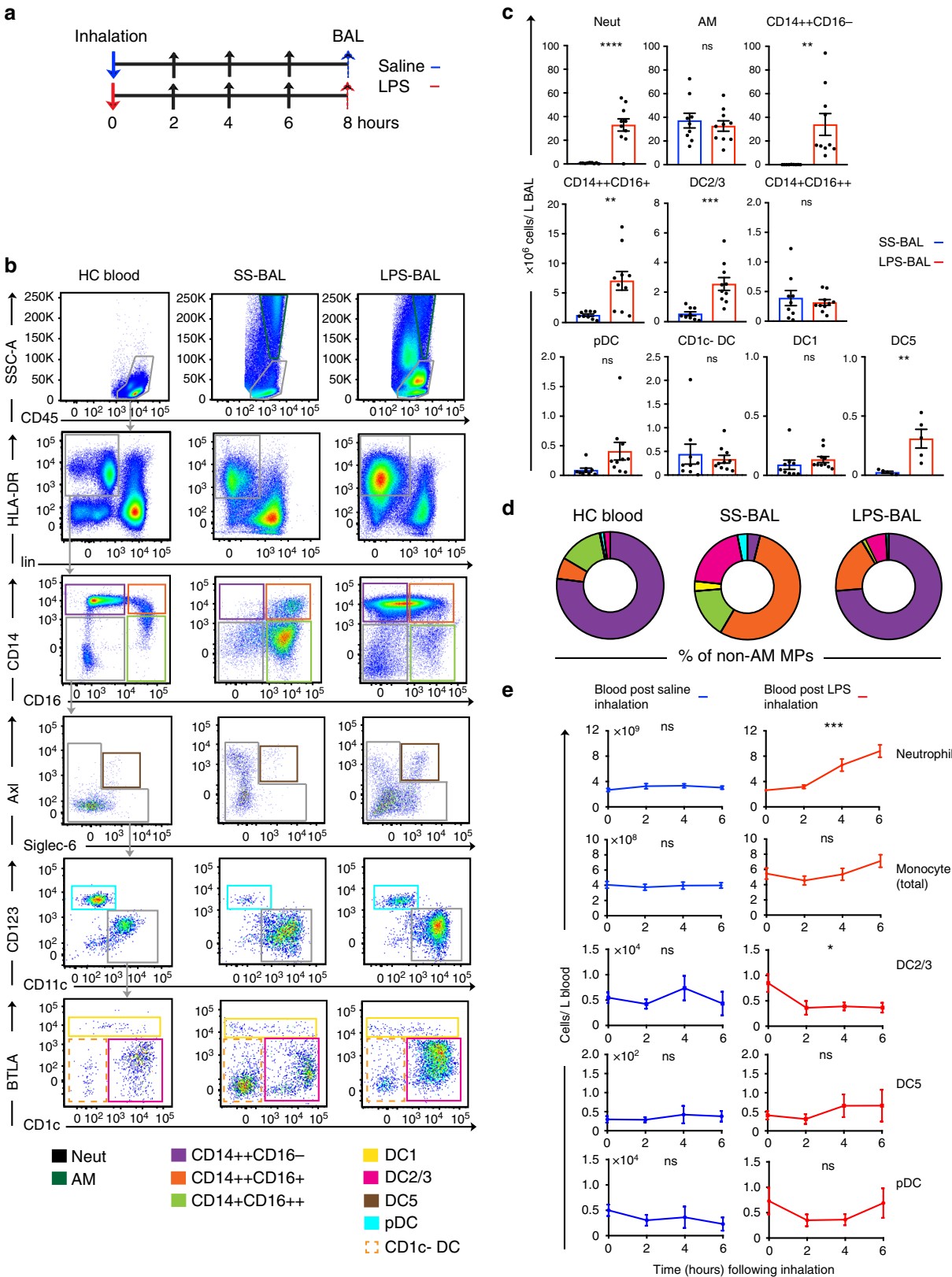

monocyte concentrations remained static. Blood DC2/3 concentration fell significantly within 2 h of LPS inhalation.

**Alveolar CD14$^{++}$CD16$^{-}$ MPs are recruited blood monocytes.** As CD14$^{++}$CD16$^{-}$ MPs were particularly enriched in LPS-BAL, we focused on this population to interrogate the molecular

changes of tissue adaptation in vivo following acute inflammatory challenge. The surface antigen phenotype of LPS-BAL CD14$^{++}$CD16$^{-}$ MPs by flow cytometry resembled that of blood monocytes. LPS-BAL CD14$^{++}$CD16$^{-}$ cells expressed genes characteristic of classical monocytes including *CD14*, *CCR2* and *SELL* but did not express *FCGR3A/B* (CD16), *CD79b*, and *CX3CR1*,

**Fig. 1** Neutrophils and mononuclear phagocytes are expanded in the alveolar airspace following LPS inhalation. **a** Schematic overview of study design. Solid arrows denote LPS inhalation (red) or saline inhalation control (blue). Black arrows denote blood sampling. Dashed arrows denote BAL. **b** Flow cytometry of leukocyte preparations from SS-BAL, LPS-BAL, and HC blood. The CD45 versus SSC plot was used to define CD45+SSC^hi AM, CD45^loSSC^mid neutrophils and CD45+SSC^lo mononuclear cells (see also Supplementary Fig. 1). Monocyte/macrophages and DCs were negative for lineage markers CD3, CD19, CD20, and CD56 and expressed HLA-DR. Monocyte/macrophages were divided into CD14++CD16−, CD14++CD16+, and CD14−CD16++ populations analogous to blood classical, intermediate and non-classical monocytes. DCs within the CD14−CD16− gate were divided into subsets: Axl+ Siglec6+ DC5s, CD11c^loCD123+ pDCs, CD1c+BTLA^lo-mid DC2/3s, and BTLA^hi DC1s. Plots are representative of $n = 9$ SS BAL and $n = 10$ LPS BAL. **c** Concentrations of neutrophil, monocyte/macrophage and DC subsets in SS-BAL and LPS BAL. Bars represent mean and lines SEM. $p$-values from unpaired $t$-tests of SS versus LPS are shown: "ns" $p > 0.05$, *$p < 0.05$, **$p < 0.01$, ***$p < 0.001$, ****$p < 0.0001$. **d** Monocyte/macrophage and DC frequency in HC blood, SS-BAL and LPS-BAL as a proportion of SSC^lo MHC class II-expressing cells (not-including CD1c- DCs). **e** Concentration of selected leukocyte populations in peripheral blood at 2-h intervals following inhalation of saline (blue line) or LPS (red line). Data points show mean ± SEM for 3–5 participants. $p$-values from one-way ANOVA are shown. $p$-value representation is described in **c**

**Table 1 Frequency of MP subsets in BAL and blood**

| MP subset | | AM | CD14++ CD16− | CD14++ CD16+ | CD14+ CD16++ | DC1 | DC2/3 | CD1c− | DC5 | pDC |
|---|---|---|---|---|---|---|---|---|---|---|
| Antigen expression by flow cytometry | CD45 | + | + | + | + | + | + | + | + | + |
| | SSC | hi | lo | lo | lo | lo | lo | lo | lo | lo |
| | Lineage | − | − | − | − | − | − | − | − | − |
| | HLA-DR | + | + | + | + | + | + | + | + | + |
| | CD14 | + | ++ | ++ | + | − | − | − | − | − |
| | CD16 | + | − | + | ++ | − | − | − | − | − |
| | CD11c | | | | | + | + | + | -/+ | − |
| | CD123 | | | | | − | − | − | -/+ | + |
| | BTLA | | | | | ++ | −/+ | − | + | + |
| | CD1c | | | | | +/− | + | − | -/+ | − |
| | Axl | | | | | +/− | +/− | +/− | + | − |
| | Siglec 6 | | | | | − | − | − | + | − |
| % of leukocytes mean (SD) | HC whole blood | 0 | 8.01 (2.04) *75.0 (7.94)* | 0.47 (0.22) *6.41 (2.29)* | 1.29 (0.85) *13.8 (6.11)* | 0.02 (0.01) *0.23 (0.22)* | 0.30 (0.1) *1.07 (0.75)* | 0.07 (0.07) *0.77 (0.72)* | 0.02 (0.02) *0.22 (0.20)* | 0.23 (0.05) *1.93 (0.93)* |
| % of MPs mean (SD) | SS BAL | 60.1 (14.3) | 1.06 (0.14) *2.92 (1.88)* | 2.38 (0.95) *43.7 (8.00)* | 0.64 (0.56) *11.4 (7.31)* | 0.12 (0.10) *2.41 (1.91)* | 0.82 (0.35) *16.1 (4.24)* | 0.69 (0.72) *12.3 (13.9)* | 0.04 (0.02) *1.06 (0.36)* | 0.13 (0.07) *2.57 (1.53)* |
| | LPS BAL | 23.1 (13.5) | 18.7 (9.13) *64.3 (17.8)* | 4.51 (3.23) *18.2 (16.6)* | 0.23 (0.16) *0.91 (0.67)* | 0.09 (0.04) *0.33 (0.15)* | 1.63 (0.61) *6.12 (2.59)* | 0.2 (0.11) *0.69 (0.37)* | 0.14 (0.05) *0.45 (0.11)* | 0.22 (0.16) *0.76 (0.49)* |

MP subsets present in BAL. MP subsets are defined by the flow cytometry gates in Fig. 1b. Expression of defining antigens is indicated as "++" bright expression, "+" positive expression, "+/−" low-level expression or "−" no expression. Where no symbol is given, expression was not measured on that subset. Frequencies of each MP subset in whole blood, SS BAL and LPS BAL are given. In standard type, frequencies are given as a percentage of total leukocytes (defined as CD45+ cells). In italic type, frequencies are given as a percentage of MPs (defined as CD45+SSC^lo, CD3,19,20,56− HLA-DR+ cells)

which are characteristic of non-classical monocytes[27] (Fig. 2a). However, unsupervised transcriptome analysis revealed distinct changes as a consequence of recruitment into the alveolar airspace.

Using the NanoString Immunology V2 579-gene panel, we compared the expression profile of HC blood monocyte subsets with SS-BAL and LPS-BAL monocyte-macrophages. By principal component analysis, the first principal component (PC1) segregated cells by tissue compartment (blood versus BAL) and the second (PC2) separated monocyte-macrophages from resident alveolar macrophages (Fig. 2b). PC3 separated the CD14++CD16− cells from SS-BAL and LPS-BAL. The distance between BAL CD14++CD16− MPs and monocytes, even by principal components 2 and 3, suggested adoption of a unique gene expression profile upon entry to the airspace. This was explored further by examination of differentially expressed genes (DEGs).

We performed pairwise comparisons of DEGs of CD14++CD16− cells between tissue compartments (HC blood vs. BAL) and between inflammatory states (SS-BAL vs. LPS-BAL). There was a core signature of 49 DEGs (11% of the 457 genes expressed in these cell types) distinguishing SS-BAL and LPS-BAL CD14++CD16− MPs from HC blood classical monocytes (Fig. 2c, Supplementary Dataset 1). The 36 up-regulated included genes encoding phagocytic receptors (MRC1, FCGR3A/B, MSR1), immunoregulatory proteins (CD274), innate immune effectors (C1QA/B), chemokines capable of monocyte and lymphocyte recruitment (CXCL10), and some genes associated with mature DC function (CD80, LAMP3). This supported the interpretation

that CD14++CD16− MPs are recruited blood monocytes with maturation and adaptation to the alveolar environment.

Comparing CD14++CD16− MPs in SS-BAL and LPS-BAL, 98 genes (21%) were differentially expressed (Fig. 2d, Supplementary Dataset 1). Steady state (SS-BAL) was associated with higher expression genes involved in antigen presentation (CD1A, HLA-DPA1, HLA-DMB, HLA-DOB, CD1D), pathogen clearance (CD209, MRC1), control of inflammatory signaling (PTPN22), and the ability to activate the allergic responses (FCEFR1A). Inflammation (LPS-BAL) saw activation of LPS response genes (IL1A, IL1B, IRAK2, TRAF3, and others) retention of monocyte-associated genes (S100A8/9, SELL), and marked expression of chemokine genes (CCL2/3/4/5/7, CXCL10, and IL8). This suggests that in quiescence, CD14++CD16− MPs have the capacity to present antigen, possibly helping to maintain tolerance, while in inflammation, they are primed to modulate immune functions through chemokine and interleukin production.

**Blood CD1c+ DC are recruited into the alveolar space.** We used a panel of antigens showing divergent expression on tissue and blood CD1c-expressing DCs (DC2/3) to evaluate the flow cytometry phenotype of CD1c+ DCs found in SS-BAL, LPS-BAL and HC blood[14,26,28] (Fig. 3a). Blood DC2/3 were CD11b^lo, Axl−, CD1a−, and CD206− while SS-BAL DC2/3 expressed all four of these antigens, confirming that these were genuine tissue DCs and not blood contaminants. However, the expression of these antigens on LPS-BAL DC2/3 paralleled that seen in blood DC2/3. Significant blood contamination of LPS-BAL was excluded by

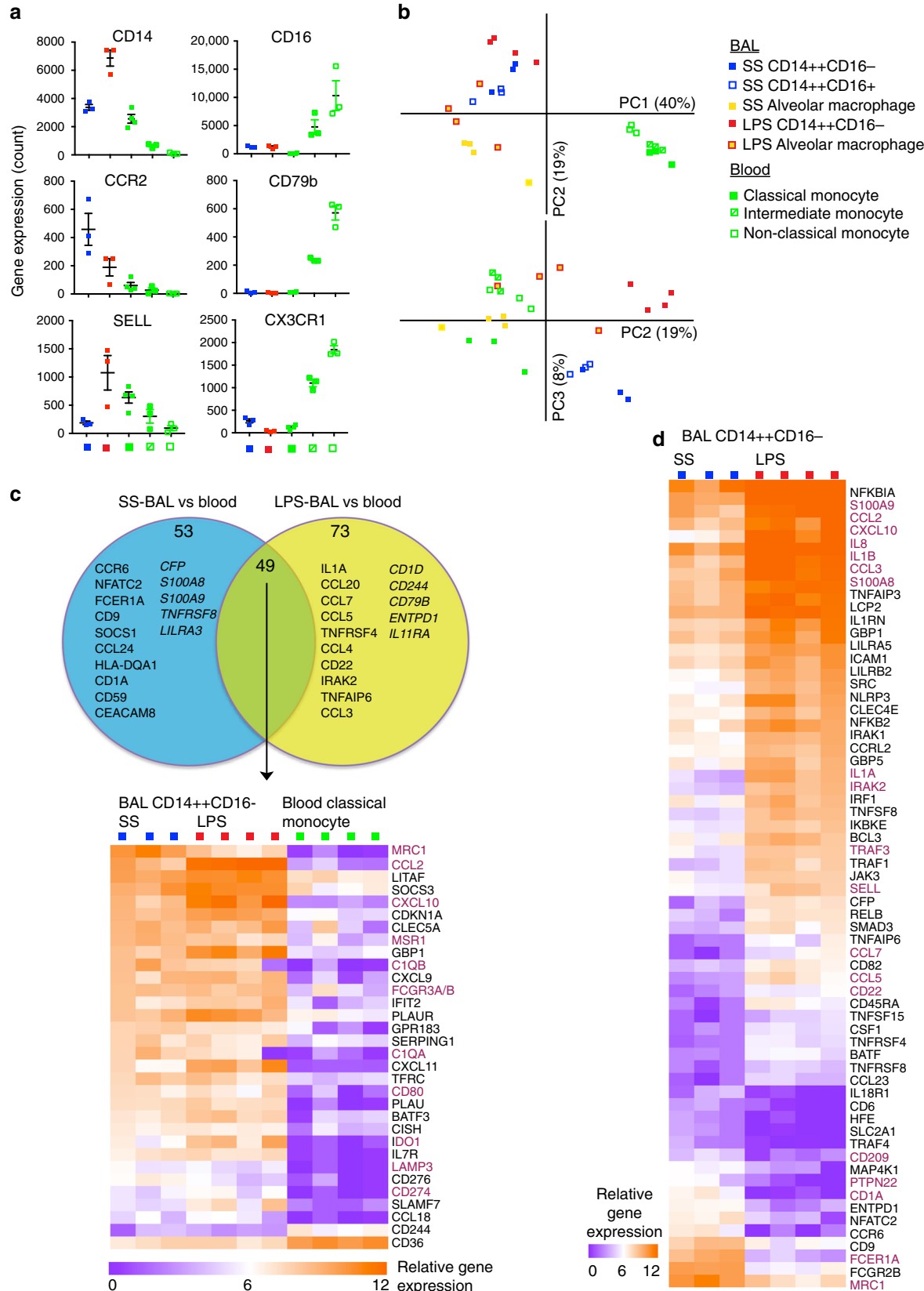

counting erythrocytes: $16.7 \times 10^6 \pm 28.4 \times 10^6$ in SS-BAL and $33.3 \times 10^6 \pm 30.8 \times 10^6$ in LPS-BAL (mean ± SD; $p = 0.04$ by unpaired $t$-test). This amounted to a 6 µl blood leak, permitting approximately 0.4% of the $10^5$ DC2/3 found in LPS-BAL to be

due to blood contaminants (DC2/3 frequency in peripheral blood is approximately 10 cells µl$^{-1}$[129]).

While the surface antigen phenotype of LPS-BAL DC2/3 by flow cytometry supported their rapid recruitment from blood

**Fig. 2** CD14$^{++}$CD16$^-$ monocyte-macrophage cells in LPS-BAL are recruited blood monocytes with transcriptional adaptations. **a** Expression of monocyte-subset discriminating genes in CD14$^{++}$CD16$^-$ cells from SS-BAL (blue square) and LPS-BAL (red square) compared with HC blood classical (filled green square), intermediate (divided green square) and non-classical (open green square) monocytes. Gene expression was quantified by NanoString array. The left-hand column contains genes with higher expression in classical than intermediate or non-classical monocytes. The right-hand column contains genes with higher expression in intermediate and non-classical than classical monocytes. Bars represent mean ± SEM. **b** Principal component analysis of immune gene expression (579-gene NanosString array) by BAL monocyte/macrophages and blood monocyte subsets. **c** Analysis of differentially expressed genes (DEGs) in BAL CD14$^{++}$CD16− cells and HC blood monocytes. Comparisons were made by unpaired $t$-test with $p < 0.05$ and >3-fold difference in mean expression. Venn diagram shows DEGs in CD14$^{++}$CD16− MPs from SS-BAL, LPS-BAL and HC blood. Circles represent DEGs between SS BAL and blood (cyan; 101 genes, 22%) and between LPS-BAL and blood (yellow; 124 genes, 27%). The overlapping circles represents DEGs shared between comparisons (49 genes, 11%). The top 10 upregulated genes (standard type) and top 5 downregulated genes (italic type) in BAL relative to blood are listed. Heatmap shows DEGs common to both SS-BAL and LPS-BAL CD14$^{++}$CD16$^-$ cells relative to HC blood classical monocytes. "Relative gene expression" refers to log2 transformed normalized gene expression count data. Genes with >10-fold difference in mean expression are shown. Genes discussed in text are colored burgundy. **d** Heatmap showing DEGs between SS-BAL and LPS-BAL CD14$^{++}$CD16$^-$ cells. Comparisons were made by unpaired $t$-test with $p < 0.05$. Genes with >5-fold difference in expression are shown. Genes discussed in the text are colored burgundy

DC2/3, consideration was also given to the possibility that they may have differentiated from recruited blood monocytes. The latter possibility was considered less likely within the eight-hour time window studied. Microarray analysis in human chronic inflammatory exudates aligned inflammatory CD1c-expressing DCs with in vitro monocyte-derived DCs[26]. From a list of conserved genes defining human and mouse monocytes and macrophages compared to DCs[28], 25 genes were evaluable on the Nanostring Immunology V2 panel. Hierarchical clustering the expression of these genes located LPS-BAL DC2/3 closest to SS-BAL DC2/3 and HC blood DC2/3 and remote from HC blood monocytes, BAL macrophages and in vitro monocyte-derived DCs (Fig. 3b). Furthermore, LPS-BAL CD1c-expressing MPs were effective stimulators of allogeneic T cell proliferation, in contrast to LPS-BAL CD14$^{++}$CD16$^-$ MPs and AMs, further supporting their origin from circulating CD1c-expressing blood DCs (Fig. 3c).

We next explored the functional adaptations to airspace residence of recruited blood DCs by comparing the immune gene expression profile of blood DC2/3 with their SS-BAL and LPS-BAL counterparts (Fig. 3d, Supplementary Dataset 1). There was a core signature of 100 DEGs in SS and LPS BAL DC2/3 compared with HC blood DC2/3 (21% of the 466 genes expressed in these cell types): 39 upregulated and 61 downregulated genes. Up-regulated genes were required for mature DC function (*CD80*, *CCR7*, *LAMP3*), involved in immunoregulation (*CISH*, *CD274*, *CD276*, *TNFRSF11A*), monocyte and T cell recruitment (*CCL2*, *CCL22*), and pathogen recognition or scavenging (*CLEC5A*, *MSR1*). The genes most down-regulated in BAL DC2/3 compared with blood DC2/3 included *CD1D*, possibly as an adaptation to the lipoprotein-rich alveolar environment, and the lymphoid-lineage associated genes *CD22 and CD244*.

Relatively few genes were differentially expressed between SS and LPS BAL DC2/3 (Fig. 3e, Supplementary Dataset 1). As with LPS CD14$^{++}$CD16−MPs, LPS DC2/3 expressed LPS response genes (*IL1B*, *IL8*, *IRAK2*) and chemokine genes (*CCL2,4,5 CXCL10,11*) to a greater extent than SS-BAL DC2/3.

**CD1c$^+$ DC heterogeneity within the alveolar space.** The flow cytometry gating strategy used for parallel examination of BAL and blood revealed heterogeneity in the CD1c-expressing DC gate by BTLA expression (Fig. 1b). In blood, BTLA expression segregates CD1c-expressing DCs into two subsets with distinct gene expression profiles, with BTLA$^+$ DC expressing lymphocyte lineage genes including *CD5, CD79A/B* and *CD24* and BTLA$^-$ DC expressing monocyte/macrophage genes such as *CD14, S100A8/9* and *F13A1* (Fig. 4a). Gene expression differences between blood BTLA$^+$ and BTLA$^-$ DC correlate with gene expression differences between blood DC2 (HLA$^-$class II genes)

and DC3 (*BST1*, *CD14*, *CD163*, *S100A8/9*)[19] (Fig. 4a) and with blood CD1c$^+$ DC subsets segregated by CD5 expression[30].

In BAL, BTLA expression discerned two subsets with the mean fluorescence intensity for BTLA$^+$ cells at $1 \times 10^3$ (corresponding to DC2) and BTLA$^-$ cells at 100 (corresponding to DC3) (Fig. 4b). The ratio of DC2:DC3 in HC blood was 40:60. In contrast, SS-BAL showed marked skewing towards DC3 (20:80). In keeping with our previous findings suggesting that LPS-BAL CD1c$^+$ DCs are recruited from circulating DCs, the DC2:DC3 ratio in LPS-BAL approximated that of blood.

The genes differentially expressed between HC blood DC2 and DC3 were not recapitulated between LPS-BAL DC2 and DC3 (Fig. 4c). The convergence in gene expression between LPS-BAL CD1c$^+$ DC subsets following recruitment to BAL may be accounted for by inflammation (*SLAMF7*, *TNFRSF9*), as well as change of compartment (*CXCL9*, *FCGR3A/B*, *S100A8/9*).

Both blood CD1c$^+$ subsets effectively stimulate allogeneic T cell production[19,30] but have distinct capacity for inducing cytokine production by CD4$^+$ T cells, with CD5$^+$CD1c$^+$ DCs (representing DC2 or BTLA$^+$ DC) inducing IL-10, IL-22, IL-4, and IL-17 production) and CD5$^{lo}$CD1c$^+$ DCs (representing DC3 or BTLA$^-$ DC) inducing IFN-γ production. We therefore compared the capacity of LPS-BAL DC2 and DC3 to influence cytokine production by CD4$^+$ T cells. We observed that both subsets were equally capable of inducing IFN-γ and IL-17 production without any IL-4 production. The capacity to induce IFN-γ production was significantly greater in LPS-BAL than SS-BAL DC2/3. Yet, even SS-BAL DC2 did not induce IL-4 production, supporting the concept that CD1c+DCs undergo functional alteration upon tissue entry (Fig. 4d).

The Axl$^+$Siglec-6$^+$ DC5 population, while enriched in LPS-BAL, remained too small to analyse in functional assays or NanoString arrays. DC5 in peripheral blood exhibit a spectrum of CD123 and CD11c expression, with CD123$^{hi}$CD11c$^{lo}$ DC5 expressing a pDC-like gene signature and CD123$^{lo}$CD11c$^{hi}$ DC5 expressing a cDC2-like gene signature[19]. SS-BAL DC5 expressed more CD123, but a greater proportion of LPS-BAL DC5 expressed CD11c (Fig. 4e). This was in keeping with a cDC2 over pDC bias in the inflamed airspace.

**Alveolar macrophages orchestrate MP recruitment.** We hypothesized that resident AMs orchestrated the recruitment of leukocytes into the alveolar airspace following LPS inhalation. LPS-BAL AMs demonstrated high expression of chemokine genes including *CCL2-4* and *CXCL10-11* (Fig. 5a, Supplementary Dataset 1). Compared with SS-BAL AMs, LPS-BAL AMs expressed 4 to 42-fold higher levels of these chemokine transcripts. Only *CXCL12* (stromal-cell derived factor) was not expressed by AMs. Corresponding protein measurements of secreted chemokines confirmed high expression of CCL2–4,

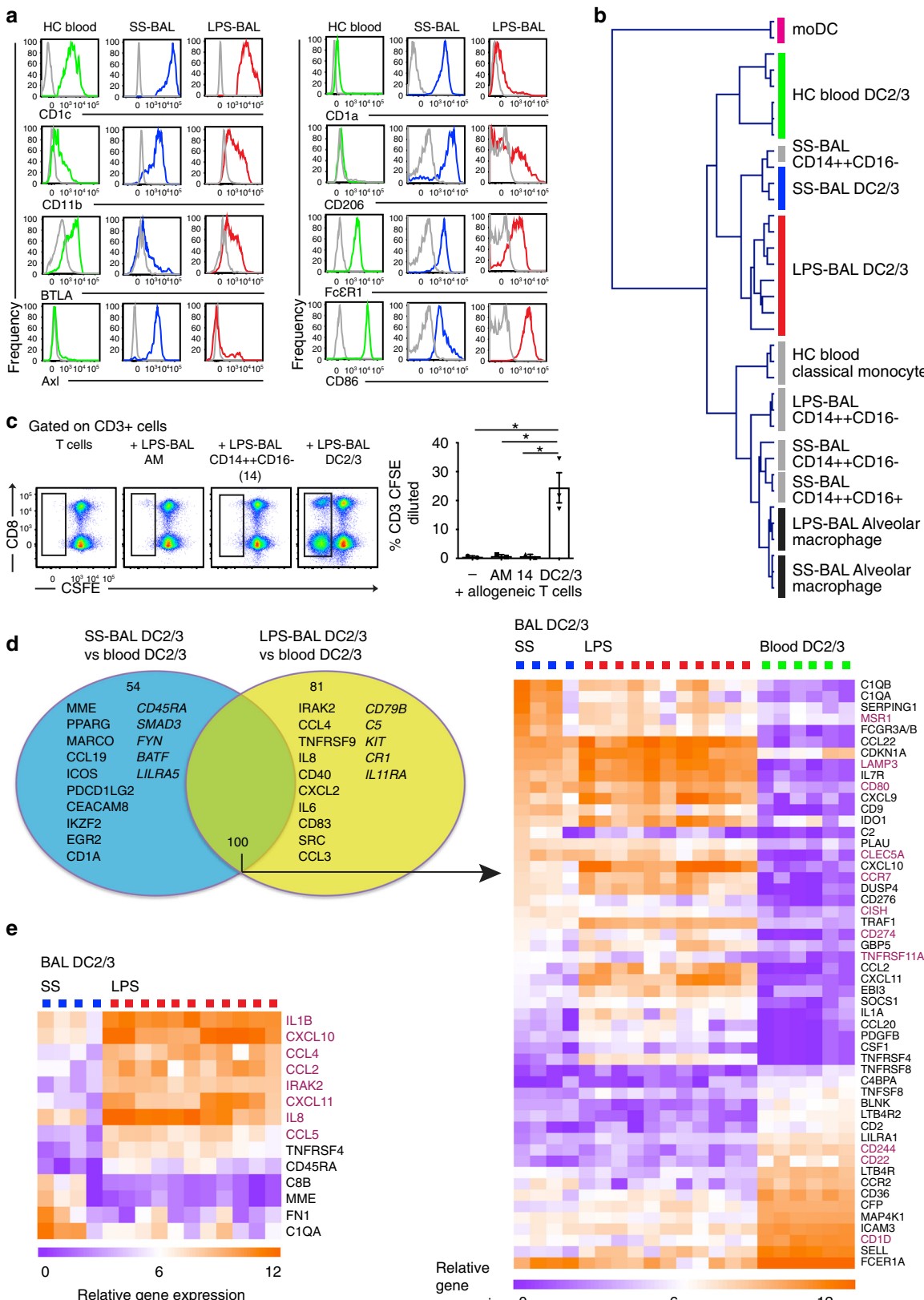

CXCL10, and CXCL12 in LPS-BAL supernatant (Fig. 5b). mRNA profiling of the cognate chemokine receptors revealed their abundance on blood monocytes and DC2/DC3 (Fig. 5c), in keeping with our findings of their recruitment into BAL following LPS challenge.

Finally, we evaluated the dynamics of pro-inflammatory cytokine secretion by resident AMs and recruited CD14$^{++}$CD16$^{-}$ monocyte–macrophages isolated from LPS-BAL. Both resident LPS-BAL AMs and recruited LPS-BAL monocyte–macrophages secreted high levels of pro-inflammatory cytokines upon re-

**Fig. 3** CD1c[+] DCs in LPS-BAL are likely to be tissue-recruited blood CD1c[+] DCs. **a** Expression of surface antigens by flow cytometry on DC2/3 from HC blood (green), SS-BAL (blue) and LPS-BAL (red) relative to isotype control (gray). Antigens predicted to discriminate between blood and tissue CD1c[+] DCs were tested. Representative plots from more than three experiments are shown. **b** Dendrogram showing hierarchical clustering of monocyte/macrophage and DC gene expression by indicated subsets isolated from HC blood and SS/LPS BAL. In vitro monocyte-derived DCs were also included. Clustering used Pearson correlation metric. Monocyte/macrophage and DC identifying genes were taken from McGovern et al.[28]. The 25 genes available on the NanoString Immunology v2 panel were used. **c** Proliferation of allogeneic peripheral blood T cells measured by CSFE dilution during 7-day co-culture with or without MP subsets isolated from LPS BAL. Flow cytometry plots are gated on CD3[+] T cells from a representative experiment. Summary graph shows 2–3 replicates per subset. Bars show mean ± SEM. Means were compared by one-way ANOVA with Dunnett's multiple comparison test of DC2/3 against other subsets. *$p < 0.05$. **d** Analysis of differentially expressed genes (DEGs) by DC2/3 from SS/LPS-BAL and HC blood. Comparisons were made by unpaired $t$-test with $p < 0.05$ and >3-fold difference in mean expression. Venn diagram shows DEGs in DC2/3 from SS-BAL, LPS-BAL, and HC blood. Circles represent DEGs between SS BAL and blood (cyan; 154 genes, 33%) and between LPS-BAL and blood (yellow; 124 genes, 39%). The overlapping circles represents DEGs shared between comparisons (100 genes, 21%). The top 10 upregulated genes (standard type) and top 5 downregulated genes (italic type) in BAL relative to blood are listed. Heatmap shows DEGs common to DC2/3 in both SS-BAL and LPS-BAL relative to HC blood. Genes with >10-fold difference in mean expression are shown. Genes discussed in the text are colored burgundy. **e** Heatmap showing DEGs between DC2/3 from SS-BAL and LPS-BAL. Genes with >10-fold difference in mean expression are shown. Genes discussed in the text are colored burgundy

stimulation with LPS in vitro, suggesting a coordinated partnership between early resident AM response which is further amplified by recruited CD14[++]CD16[−] monocyte–macrophages in mediating early acute tissue inflammation (Fig. 5d).

## Discussion

The human MP system is a complex repertoire of innate immune cells distributed across distinct anatomical compartments that serve heterogeneous functions in the inflammatory response. Unraveling this complexity is necessary to understand tissue immune surveillance, immunopathology and inform vaccine design.

In keeping with previous reports[15,31,32], we identified alveolar airspace DC subsets (DC1, DC2/DC3, and pDC) corresponding to those previously described in peripheral blood. We established that BTLA distinguishes two CD1c[+] DC subsets in BAL with equivalent gene expression profiles to DC2/CD5[+] DCs and DC3/CD5[lo] DCs in blood[19,33]. Separate CD1c[+] subsets have also been described in skin and arise independently from haematopoietic precursors, confirming that these are stable subsets and not simply variations in activation state[33]. We showed that BTLA expression is another reliable marker of DC1 in blood and lung and verified the identity of sorted BTLA[hi] cells through their expression of DC1 defining genes (e.g., XCR1, TLR3, and IRF8). Two other studies[14,18] did not identify DC1 in lung interstitial tissue, which may be explained by the experimental design and gating strategy used for analysis. We identified a distinct pDC population in BAL, consistent with some reports[15,17,34] but not others[18]. The inconsistent finding of pDC in BAL does not arise from their selective vulnerability to cryopreservation[17]. Although Desch et al. did not find a CD123+CD303+pDC population in healthy lung tissue, CD303+ cells were identified in tumor-bearing lung[14], possibly indicating that pDCs in lung are restricted to the alveolar airspace and only present in parenchyma during pathological situations.

By mapping alveolar airspace MP subsets relative to blood populations, we were able to compare MP populations across compartments upon LPS-delivery into the airway. In most previous studies, tissue inflammatory composition has been examined in isolation without reference to blood and origins have been inferred[35–37]. Due to our referencing of blood it appeared logical that CD14[++]CD16[−] MPs infiltrating BAL would be monocytes recruited from blood with tissue adaptation, but the nature of tissue adaptation required investigation. In both mouse and human studies, monocytes recruited to tissues can acquire DC or macrophage characteristics[20,22,26]. Monocytes have also been described entering tissue with minimal adaptation[38]. The type of MP characteristics adopted is highly relevant to the subsequent

immune response as macrophages and DCs differ in capacity for migration, phagocytosis, ability to induce adaptive immune responses and in the cytokine/chemokine secretion profile. Through analysis of immune gene expression we established that recruited alveolar CD14[++]CD16[−] cells in LPS-BAL and their counterpart in SS-BAL were distinct from blood monocytes. They expressed a broad array of genes important for mature macrophage function including phagocytic receptors, immunoregulatory proteins and cytokines. Despite the LPS activation signature present in recruited cells, there was commonality in the gene expression profiles of steady state and inflammatory CD14[++]CD16[−] cells, emphasizing the importance and rapidity of the impact of tissue microenvironment on shaping monocyte fate following extravasation.

In addition to the expansion of monocyte–macrophages, we identified several DC subsets expanded in the alveolar airspace. Recruitment of DC from the blood to the gas exchanging regions of the previously healthy human lung during early inflammation has never been documented before. This recruitment was observed for specific DC subsets, including CD1c-expressing DC2/3 and Axl[+]Siglec6[+] DC5 but not DC1 and pDCs. Sequential analysis of peripheral blood DC2/3 following LPS inhalation detected significant depletion during the period of accumulation in BAL. Analysis of surface antigens with differential expression between blood and tissue demonstrated that recruited DC had an antigen profile comparable with blood, consistent with recruitment of blood DCs and providing evidence against translocation of DCs from the lung interstitium. In the most comprehensive description of human inflammatory DCs to date, CD1c-expressing DCs in chronic inflammatory exudates were transcriptionally aligned to in vitro monocyte-derived DCs[26]. This fits with numerous observations in mouse that monocyte-derived cells accumulate in tissues under inflammatory conditions and can adopt DC characteristics[3,5,20,23,25]. In vivo data suggests it takes at least 24–48 h for monocytes to differentiate into DCs[39] and in vitro, generation of DCs from monocytes (human) or bone-marrow derived cells (mouse) takes 5–7 days[40,41]. In view of our 8-h time-course, the rapid accumulation of CD1c[+] DCs into alveolar airspace following LPS inhalation is most in keeping with recruitment from blood and is consistent with previous observations of CD1c[+] DCs accumulating in bronchial (conducting airway) mucosa within 4 h of allergen challenge[35], though detailed analyses of the infiltrating population was not possible in this earlier study. Transcriptome analysis further distinguished in vitro monocyte-derived DCs from LPS-BAL DCs. Our data suggest that in acute tissue inflammation, CD1c[+] DCs, including a subset with shared expression of monocyte genes, can be directly recruited from blood and arise independently of monocyte-

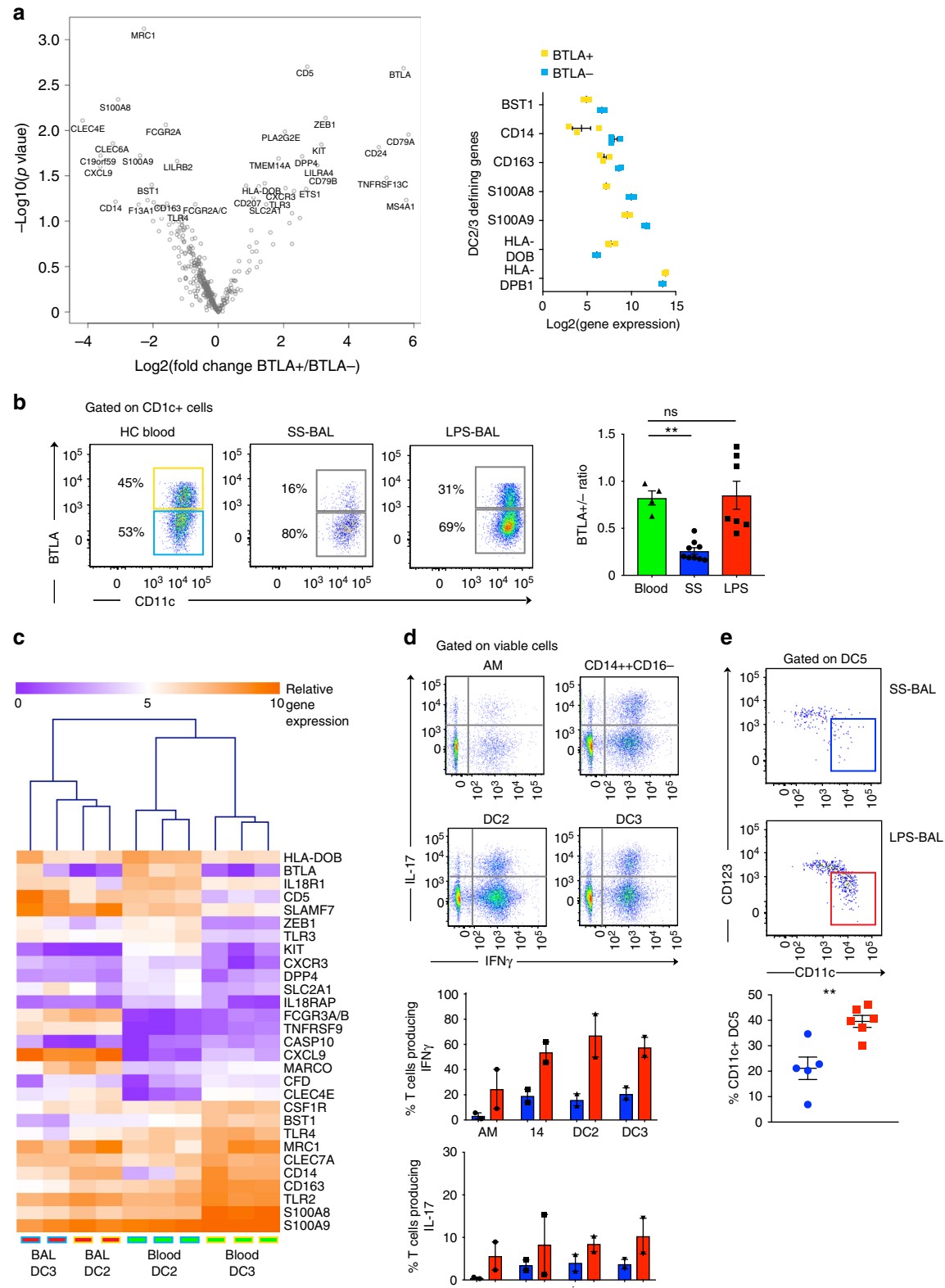

differentiation. This finding is likely to reflect the early time-point we were able to study in our experimental model. Later characterization of BAL following experimental inflammatory stimuli and analysis of chronic disease states will reveal whether monocyte-derived DCs arise subsequent to this initial inflammatory DC influx.

We propose that resident AMs operate to recruit DCs into the alveolar airspace, in addition to their role in recruiting neutrophils and monocytes. Activation of AMs by LPS resulted in rapid upregulation of chemokines involved in leukocyte migration and pro-inflammatory cytokine mediators, elucidating how robust neutrophil recruitment and lung inflammation can be

**Fig. 4** The pool of recruited LPS-BAL CD1c⁺ DCs contains DC2 and DC3, which are functionally altered on entry to the inflamed airspace. **a** Volcano plot shows DEGs between BTLA⁺ and BTLA⁻ CD1c⁺ DCs from HC blood. DEGs were calculated by unpaired $t$-test with $p$-value adjustment using Benjamini-Yekutieli method. Genes with $p < 0.05$ are displayed as text. Accompanying scatterplots verifiy that that blood BTLA⁺ and BTLA⁻ CD1c⁺ DCs exhibit the expected gene expression profile of DC2 and DC3 respectively. Plots show expression from $n = 3$ samples with bars showing mean ± SEM. **b** Comparison of BTLA⁺:BTLA⁻ ratio in CD1c⁺ DCs from HC blood, SS-BAL and LPS-BAL by flow cytometry. The gating strategy represented in Fig. 1b was used to define CD1c⁺ cells. Flow cytometry plots show percentage of BTLA⁺ and BTLA⁻ CD1c⁺ cells in representative examples of blood, SS-BAL and LPS-BAL. Accompanying bar graph summarizes flow cytometry data from $n = 4$–9 replicates. Statistical comparison by one-way ANOVA with Dunnett's multiple comparisons tests. ns $p > 0.05$, **$p < 0.01$. **c** Heatmap showing expression of the 29 genes with differential expression between blood DC2 and DC3 ($p < 0.05$ and fold difference >2) in DC2 and DC3 sorted from LPS-BAL and HC blood. Samples were clustered using Euclidian distance metric. **d** Flow cytometry read-out of a T cell cytokine production assay following 10-day co-culture of allogeneic peripheral blood CD4⁺ T cells with macrophages and DCs isolated from LPS-BAL. Accompanying bar graphs summarize flow cytometry data showing mean ± SEM IFN-γ and IL-17 production following co-culture of SS-BAL (blue bars) or LPS-BAL (red bars) macrophages and DCs (each $n = 2$). Two-way ANOVA of IFN-γ production gives ***$p < 0.001$ for SS-BAL vs. LPS-BAL and ns $p = 0.07$ for MP type. Two-way ANOVA of IL-17 production gives ns $p = 0.06$ for SS-BAL vs. LPS-BAL and ns $p = 0.67$ for MP type. **e** Flow cytometry plots show representative examples of CD123 and CD11c expression by Axl⁺Siglec6⁺ DC5 in SS-BAL and LPS-BAL. The gating strategy represented in Fig. 1b was used to define DC5. Gates demarcate CD11c-expressing cells. Accompanying scatterplot summarizes flow cytometry data, showing proportions of DC5 expressing CD11c in SS-BAL (blue circles) and LPS-BAL (red squares). **$p < 0.01$ by unpaired $t$-test

observed even after monocyte-depletion in a human LPS inhalation model[42]. The acute inflammatory cascade initiated by resident AMs is further amplified by recruited $CD14^{++}CD16^{-}$ monocyte-macrophages.

There are a number of limitations to this study. LPS inhalation provides the opportunity to study early time points in inflammation but it is not an accurate representation of human disease. Infective stimuli are rarely presented to the immune system as single bolus and will frequently be accompanied by other pathogen-associated molecular patterns. BAL permits effective sampling of the alveolar space, but "leaves behind" alveolar epithelial cells, adherent infiltrating cells, and those migrating through the interstitium. These populations could make important contributions to the regulation of the inflammatory response but cannot be sampled without recourse to biopsies that are unlikely to be ethical in healthy volunteers and may be subject to sampling error. Furthermore, BAL itself is inflammatory to the alveolar region, limiting the opportunity to study serial cell migration in the same individual, or to assess "recovery" BAL samples in this setting.

Our study has permitted detailed phenotypic, transcriptional, and functional analyses of MPs present in the alveolar airspace of healthy volunteers inhaling saline control or LPS to induce acute local inflammation. In LPS-BAL, we observed perturbation of the MP profile, noting significant influx of $CD14^{++}CD16^{-}$ monocyte–macrophages and selected DC subsets, likely regulated by resident AMs. This finding adds to our understanding of the potential role of blood DCs and the possibility of monocyte-independent inflammatory DCs at early time points in acute inflammation. Understanding the in vivo kinetics and dynamic regulation of MP following local LPS stimulation extends biological insights into the mechanisms of acute inflammation. Dissection of the functional consequences for the host will provide the opportunity to understand both beneficial and detrimental effects of such inflammation.

## Methods

**Ethical approval.** The LPS inhalation study was approved by Newcastle & North Tyneside 2 Research Ethics Committee of the NHS Health Research Authority (REC reference number 12/NE/0196) under the full title "A lipopolysaccharide (LPS) inhalation model to characterize divergent cellular innate immune responses and presence of alveolar leak early in the course of acute lung inflammation". The study was conducted according to the principles expressed in the Helsinki Declaration and informed consent was obtained from all participants.

**Study population.** The study was advertised to university undergraduates. Potential recruits were invited to a screening interview to assess their suitability for participation. Inclusion criteria were healthy adult volunteers aged 18 to 40 able to give informed consent. Exclusion criteria were age <18 or >40 years; history of chronic respiratory disease, diabetes, heart disease, renal disease or recurrent infections; current respiratory tract infection; taking prescription medication (except oral contraceptives); current smoking or history of smoking 20 cigarettes per day for more than 2 years or any smoking in the past year; alcohol intake of more than 21 units per week; pregnancy or lactation; abnormal examination findings at screening (temperature >37.3 °C or oxygen saturation <95% breathing room air); abnormal blood results at screening (hemoglobin concentration, total white cell count or neutrophil count outside the gender-specific laboratory reference ranges; platelet count $<100 \times 10^9 l^{-1}$; serum sodium, potassium, creatinine or alanine aminotransferase outside laboratory reference ranges; serum urea >10 mg $dl^{-1}$; serum bilirubin >30 $\mu$mol $l^{-1}$); abnormal spirometry at screening (FEV1 or FVC <80% predicted or FEV1:FVC ratio <70%). Eligible volunteers were given verbal and written information about the study and at least 24 h to consider their participation before signing a consent form. All primary research documents were anonymised and participant details kept confidentially in accordance with Caldicott guidelines. Data from 19 participants are presented ($n = 10$ LPS, $n = 9$ saline).

**Study interventions.** Volunteers were allocated to inhale 54 μl sterile 0.9% sodium chloride with or without 60 μg LPS from *E. coli* 026:B6 (Sigma). Participants were allocated sequentially to receive saline or LPS to give optimal control over downstream experiments and they were not made aware of which intervention they had received. Delivery was targeted at the lower airways using an automatic inhalation-synchronized dosimeter nebulizer (Spira, Hameenlinna, Finland). The test solution was released following inhalation of 50 ml air to ensure that laminar flow was established. Participants performed a 5 s breath hold at vital capacity to promote deposition of LPS in the lower respiratory tract.

Participants were asked about symptoms (flu-like symptoms, sore throat, cough, wheeze, chest pain, sputum production, nasal secretions, or any other symptom) immediately after inhalation and at 6 h. Body temperature was measured hourly until 6 h. Venous blood samples were obtained at 0, 2, 4, 6, and 24 h after inhalation. We used blood leukocyte counts as the indicator of LPS effect.

Flexible fiber-optic bronchoscopy was performed 8 h after inhalation. Intravenous sedation with midazolam was available but all participants elected for a non-sedated procedure. Participants received topical administration of 1% lignocaine spray to the mouth and pharynx. Bronchial wash of the upper airways was performed with 20 ml warmed 0.9% sodium chloride and discarded. BAL of the right middle lobe was performed with 150 ml warmed 0.9% sodium chloride and retrieved by gentle suction.

**Participant safety.** History, bedside observations, cardio-respiratory examination, and spirometry were performed immediately before LPS or saline inhalation. Bedside observations were repeated hourly and spirometry repeated at 6 h after inhalation. If FEV1 fell by 10% from baseline, bronchoscopy was canceled. Prior to bronchoscopy, participants were fasted for 4 h. There was constant monitoring of oxygen saturations and electrocardiogram during bronchoscopy. Patients were observed for 30–60 min after bronchoscopy and allowed to leave if bedside observations and cardio-respiratory examinations were normal. Written and verbal advice was given to avoid eating and drinking within two hours of local anesthetic administered to the mouth and pharynx. Participants were informed that LPS inhalation and bronchoscopy may result in temperature, mild headache, shivering, dry cough, and upper airway discomfort.

**Cell isolation.** Blood was collected into EDTA. PBMC were isolated by density centrifugation using Lymphoprep (Stemcell technologies) according to manufacturers instructions. BAL samples were kept at room temperature for 60–90 min before processing at 4 °C. BAL fluid was passed through a 100 μ filter (Falcon). Following centrifugation, cell-free supernatant was stored at −80 °C and cells were prepared for immediate analytical flow cytometry and cell sorting.

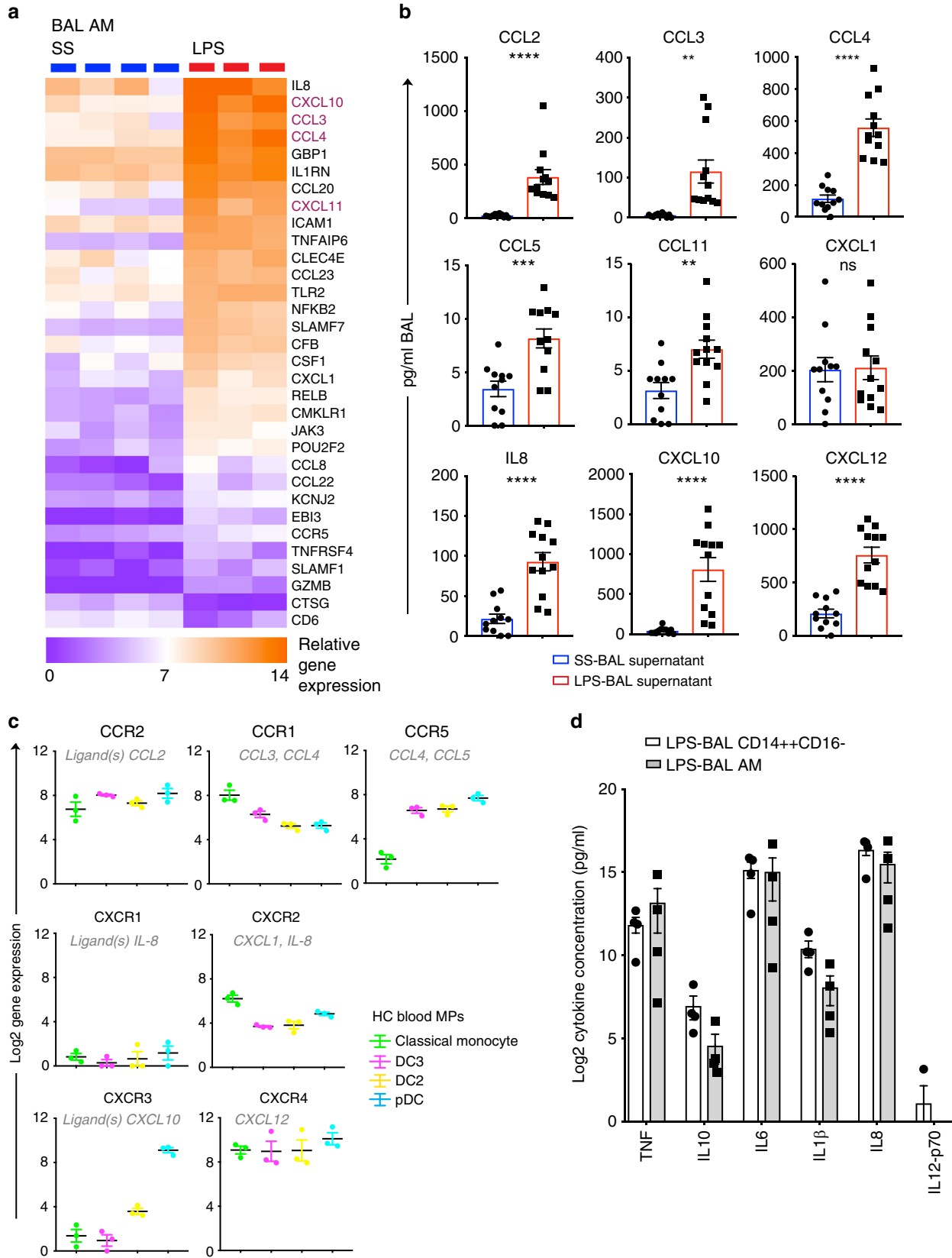

**Flow cytometry and sorting**. Cell pellets were washed in PBS (Sigma) supplemented with 2% fetal calf serum (Sera lab) and 2 mM EDTA (Invitrogen). The following antibodies were applied for 20 min at 4 ℃ (clone; supplier; catalog number; final concentration): anti-CD45-V500 (HI30; BD; 560777; 1:25), anti-CD3-FITC (SK7; BD; 345763; 1:25), anti-CD19-FITC (AG7; BD; 347543; 1:25), anti-CD20-FITC (L27; BD; 345792; 1:25), anti-CD56⁻FITC (NCAM16; BD; 345811; 1:25), anti-HLA-DR-BV786 (G46⁻6; BD; 307642; 1:25); anti-CD14⁻BV650 (M5E2; Biolegend; 301835; 1:40), anti-CD16-PE/Dazzle-594 (3G8; Biolegend; 302054; 1:33), anti-CD11c-APCCy7 (Bu15; Biolegend; 337218; 1:33), anti-CD123-PERCPCy5.5 (7G3; BD; 558714; 1:25), anti-CD1c-PECy7 (L161; Biolegend;

**Fig. 5** Alveolar macrophages recruit monocytes and DCs and secrete pro-inflammatory cytokines upon acute LPS challenge in vivo. **a** Heatmap of immune genes showing differential expression between AM isolated from saline BAL and LPS BAL. Genes with >5 fold difference in expression and $p < 0.05$ are shown. Genes discussed in the text are colored burgundy. **b** Quantification of chemokines in SS-BAL (blue bars) and LPS-BAL (red bars) supernatant by multiplexed ELISA (Luminex). Bars show mean ± SEM. By unpaired $t$-test, ns, not significant, **$p < 0.01$, ***$p < 0.001$, ****$p < 0.0001$. **c** Chemokine receptor gene expression on HC blood classical monocytes (green), DC2 (yellow), DC3 (pink) and pDC (cyan) quantified by Nanostring. Bars show mean ± SEM of $n = 3$ for each subset. Log2 gene expression is shown. Ligands for each receptor are listed in gray italic type. **d** Inflammatory cytokine production by CD14$^{++}$CD16$^-$ cells and AM retrieved from LPS BAL (each $n = 4$) and stimulated ex-vivo with LPS 100 ng ml$^{-1}$ for 10 h. Cytokines were measured by cytometric bead array. Bars show mean ± SEM. CD14$^{++}$CD16$^-$ cells were compared with AM by $t$-tests with Holm-Sidak multiple comparison correction and alpha < 0.05. No comparisons were significant

331506; 1:40), anti-CD11b-APC (ICRF44; Biolegend; 301309; 1:25), anti-BTLA-PE (J168–540; BD; 558485; 1:100), anti-Axl-APC (108724; R&D Systems; FAB154A: 1:33) and anti-Siglec-6-A700 (767329; R&D Systems; FAB2859N; 1:33). CD1c$^+$ DC phenotyping experiments additionally used anti-CD1a-AF700 (HI149; Biolegend; 300120; 1:50), anti-CD206-PE (19.2; BD; 555954; 1:25), anti-FcER1-PE (CRA1; eBioscience; 12–5899–42; 1:25) and anti-CD86-PE (FUN-1; BD; 555658; 1:25). Dead cells were excluded with DAPI (Partec). Analytical flow cytometry was performed on a BD Fortessa X20 or BD FACSCanto II and cell sorting with a BD FACS Fusion running FACSDiva version 7. Consistent instrument performance was ensured by running Cytometer Setup and Tracking Beads (BD). PBMC were periodically run through the analysis template as a biological control to ensure gates were capturing the expected populations. Doublets were excluded with a SSC-H vs. SSC-A plot. FlowJo version 9.6.7 was used for analysis. Sorting strategies are given in Supplementary Fig. 3.

**Peripheral blood MP quantitation**. Quantitation of neutrophils and total monocytes was by complete blood count (CBC) in a UKAS accredited clinical laboratory using the Sysmex XE-2100. For MP subsets not quantitated in the CBC, 1 ml whole blood was lysed using an in-house ammonium chloride lysis buffer and transferred to a Trucount tube (BD). Antibodies were applied as above and flow cytometry performed following a final red cell lysis step. Data were acquired on a BD Fortessa and cell concentrations calculated with reference to bead counts.

**Chemokine/cytokine analysis**. Chemokines and cytokines in BAL fluid supernatant were quantified using multiplexed ELISA (ProcartaPlexTM 34−plex, EBioscience). Captured analytes were detected on a Qiagen Liquichip 200 running Luminex 100 integrated system software version 2.3. Procartaplex Analyst version 1.0 was used to define standard curves and determine analyte concentrations.

**Gene expression analysis**. Lysates from $1–2 \times 10^4$ cells per sorted population were prepared using 5μl buffer RLT (Qiagen) with 1% beta-mercaptoethanol (Sigma). Transcripts from 579 immunology genes were detected using a Nano-String Immunology v2 panel according to the manufacturers instructions. Experiments using blood and BAL DC2/3 included an additional 30-gene add-on to the Immunology v2 panel, designed to detect DC and monocyte/macrophage genes, including *ASIP, DBN1, MERTK, C19orf59, F13A1, NDRG2, CCL17, FGD6, PACSIN1, CD1C, FLT3, PPM1N, CD207, GCSAM, PRAM1, CLEC10A, GGT5, S100A12, CLEC9A, Ki67, SIRPA, CLNK, LPAR2, TMEM14A, COBLL1, LYVE1, UPK3A, CXCL5, MAFF, ZBTB46*. Panels included 15 housekeeping genes. Data normalization was performed in nSolver version 3, including a background subtraction step, positive control normalization using synthetic spike-in controls and content normalization using a combination of 15 housekeeping genes.

**T cell proliferation assay**. T cells were isolated by negative selection from healthy control peripheral blood collected into EDTA using RosetteSep Human T Cell Enrichment Cocktail and Lymphoprep (Stemcell technologies). Aliquots were cryopreserved at −80 °C. Thawed T cells were labeled with 1 μM CSFE (Invitrogen) prior to co-culture. MP populations were sorted from BAL and co-cultured with T cells at a 1:25 ratio in 96 well v-bottom plates containing RPMI 1640 (Lonza) with 100 U ml$^{-1}$ penicillin, 10 μg ml$^{-1}$ streptomycin, 2mM ʟ-glutamine (all Invitrogen) and 10% heat-inactivated fetal calf serum (Sera Lab). Cultures were maintained for 7 days at 37 °C and 5% CO$_2$. Outputs were prepared for flow cytometry as described above using anti-CD3-V500 (UCHT1; BD; 561416; 1:50), anti-CD4-PE (RTPA-T4; BD; 555347; 1:50), anti-CD8-APCCy7 (SK1; BD; 557834; 1:50) and analyzed on a FACS Canto II running FACSDIVA version 7. Three experiments were performed on populations sorted from LPS BAL.

**T cell cytokine production assay**. CD4 T cells were isolated by negative selection from healthy control peripheral blood collected into EDTA using RosetteSep Human CD4 T cell enrichment cocktail. Aliquots were cryopreserved at −80 °C. MP populations were sorted from SS-BAL or LPS-BAL and co-cultured with T cells at a 33:1 ratio in 96-well round-bottom plates containing RF10. On day 6, medium was replenished with RPMI 1640 with supplements as above and 10 U ml$^{-1}$ human recombinant IL-2 (Immunotools). On day 10, cells were stimulated with 10 ng ml$^{-1}$ phorbol 12-myristate 13-acetate (Sigma) and 1 μg ml$^{-1}$ ionomycin (Sigma) for 4 h, with 10 with brefeldin A (Sigma) added after the first hour. Dead

cells were stained with a Zombie Aqua (Biolegend) prior to fixation and permeabilization (BD) according to manufacturers instructions. Cells were prepared for flow cytometry as above using anti-IFN-γ-PE/Dazzle-594 (4s.B3; Biolegend; 505845; 1:100), anti-IL17-AF647 (BL168; Biolegend; 512309; 1:100) and anti-IL4-PECy7 (MP4–25D2; Biolegend; 500823; 1:100) and analyzed on a Fortessa X20 (BD) running FACSDIVA version 7.

**Ex-vivo macrophage stimulation**. Alveolar macrophages and CD14$^{++}$CD16$^-$ monocyte/macrophages were sorted from LPS BAL ($n = 4$ each). $1 \times 10^5$ sorted cells were cultured in 96-well round bottom plates in 100 μl RPMI 1640 with supplements (as above) containing 100 ng ml$^{-1}$ LPS from *E. coli* (Sigma). After 10 h, supernatants were harvested and stored at −80 °C. Supernatants were batch-analysed by cytometric bead array (BD) to detect IL-10, IL-1β, TNF, IL-6, and IL-8. Bead populations were resolved on a FACS Canto II running FACSDIVA Version 7 and analyzed using FCAP Array software version 3.

**Generation of monocyte-derived dendritic cells (moDC)**. Classical monocytes were isolated from healthy control peripheral blood by density centrifugation and FACS sorting. $5 \times 10^5$ monocytes were cultured in 500 μl RPMI 1640 with supplements (as above) in 24-well plates for 5 days at 37 °C and 5% CO$_2$. Medium contained 50 ng ml$^{-1}$ GM-CSF (R&D) and 50 ng ml$^{-1}$ IL-4 (Immunotools). Medium and cytokines were refreshed on day 3. Cells were harvested and FACS-sorted on day 5, to exclude undifferentiated CD14$^+$ monocytes and include only CD1c$^+$ moDC.

**Statistical analysis**. Statistical analyses stated in the text were performed using GraphPad Prism version 7.0. Gene expression analyses were performed in Multi-Experiment Viewer version 10.2 and nSolver v4 advanced analysis module.

## Data availability

NanoString gene expression data have been deposited in Gene Expression Omnibus (GEO) with the accession code GSE126923. Other data that support the findings of this study are available from the corresponding author upon reasonable request.

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

## Acknowledgements

This work was funded by Wellcome Trust Clinical Research Training Fellowship WT097941 (L.J.) with additional contribution from the Newcastle upon Tyne Hospitals Healthcare Charity and the NIHR Newcastle Biomedical Research Center (A.J.S.). We would like to thank the Clinical Research Facility of the Newcastle upon Tyne Hospitals NHS Foundation Trust and Newcastle University and the Endoscopy Unit at the Royal Victoria Infirmary, Newcastle. We appreciate and acknowledge support from the Newcastle University NanoString Service and Dr Venetia Bigley, and the Newcastle University Flow Cytometry Core Facility.

## Author contributions

L.J., S.W., G.R., D.M., A.Fu, K.G., A.F., I.F., M.H.R.S., J.S., M.C., M.H., and A.J.S. Conceptualization: A.J.S., M.H., M.C., L.J. Methodology: A.J.S., S.W., D.M., A.Fu, K.G., A.F., M.C., M.H., A.J.S., L.J. Investigation: S.W., L.J., G.R., I.F., M.H.R.S., J.S. Writing-original draft: L.J., A.J.S., M.H. Writing-revision and editing: LJ, AJS, MH Funding acquisition: A.J.S., L.J. Resources: A.J.S., M.H., M.C. Supervision: A.J.S., M.H., M.C.

## Additional information

**Competing interests:** The authors declare no competing interests.

