## [Peer Review File · Nature Communications]

Reviewers' comments:

Reviewer #1 (Remarks to the Author):

Acute lipopolysaccharide inhalation recruits dendritic cells and monocytes to the alveolar airspace

This is an excellent paper on recruited monocytes and DCs rapidly adopted gene expression profiles following lipopolysaccharide inhalation.

I feel sorry to say that I am a clinician who was involved in numerous LPS inhalation studies but I am not an expert on monocytes and DCs. Taken this into account I can only comment on the clinical data provided in the paper. The lab findings have to be reviewed by an expert on DCs and MPs.

Clinical review:

60 µg LPS is in the dose range of many studies and after 8 hours some clinical and laboratory findings are often recorded. Unfortunately these data are not provided by the authors. It would be helpful if the authors provide data on clinical symptoms after LPS inhalation at least in the supplement.

The following data should be shown:

What side effects (cough and flue like symptoms) did you record?

Please record data on the increase of body temperature? It is well known that after LPS challenge there is an increase of at least 0.5 celsius after 8 hours.

In addition please provide a figure on leukocyte- and CRP increase after 8 hours and if available after 24 hours to compare the inflammatory response given to previous studies.

Since this is a clinical studies with human volunteers the study has to be registered for instance by clinicaltrial.gov. Unfortunately no clinical trial number is provided-.why?

No data on the power calculation, primary and secondary parameters are provided-why? Why did you challenge ten and not 16 patients?

I recommend that the authors show the complete study protocol in the supplement as it is usually done in consort journals.

Reviewer #2 (Remarks to the Author):

1. The investigators have provided data on the effect of LPS versus saline (control) inhalation in healthy subjects done 8 hours after exposure with bronchoscopy and BAL 8 hours. The purpose was to characterize the mononuclear phagocytes recruited to the air spaces after LPS inhalation. Classical monocytes and two DC subsets were expanded by LPS. The DCs appear to be recruited from the circulating blood. The recruited MPs acquired gene expression profiles that are characteristics of alveolar macrophages. The number of chemoattracted MPs to the lung was a small fraction of circulating MPs so it cannot be determined whether the source was spleen, liver, or bone marrow or just the blood. There was considerable care and effort to do these studies.

2. Although these studies provide new information in the human lung beyond just mouse lung

studies, the significance of the research would be greater if they were compared to patients who developed acute bacterial pneumonia. The authors in fact do state in their discussion that their objective is to understand the function of recruited MPs and other cells in the setting of inflammation. LPS is a way to simulate early inflammation but why not study a set of patient with early acute pneumonia for comparison?

3. The authors close the discussion with the theory that this data can identify potential anti-inflammatory targets, but what is the rationale for impeding or blocking any of these pathways? Are they part of necessary host defense in the presence of pathogen induced pneumonia. If the authors are thinking about ARDS, they should be cautious here as these host responses have evolved to probably enhance host defense.

4. The manuscript does lack a good paragraph on limitations of these studies.

5. I suppose it was too difficult to obtain consent for a later BAL, but a follow up BAL at 24-48 hours would have added more time-dependent information.

Reviewer #3 (Remarks to the Author):

Comments for Authors

The goal of this study was to better understand the subsets of mononuclear cells recruited to the alveolar spaces of human lungs in response to a defined inflammatory stimulus, bacterial lipopolysaccharides (LPS). The authors used a single stimulus that activates the TLR4 pathway and assessed inflammation in the lungs of humans by bronchoalveolar lavage (BAL) at one time (8 hrs) after inhalation of LPS. Blood and lung cells were analyzed by multiparameter flow cytometry and cell sorting, and gene expression in mononuclear cell subsets was analyzed by mRNA analysis. Inhaled sodium chloride solution was used as a comparison for inhaled LPS. The findings are directly relevant to lung inflammation in humans in response to a defined bacterial stimulus.

1. Overall, this is a descriptive study that characterizes mononuclear cell populations in the lungs of human volunteers at a single time following the inhalation of a defined stimulus (gram negative LPS, TLR4 agonist). Some of the results are new (e.g. the findings related to DC subpopulations), but other aspects are predictable based on existing literature (e.g. that blood MN migrate into the lungs and modify gene expression and that alveolar macrophages contribute to leukocyte recruitment in response to LPS). The authors need to be more clear about how this study advances the understanding of lung inflammatory responses in humans. In addition, do the lung responses appear to differ from what is known in other tissues?

2. Table 1. A footnote should be added to the Table to define what the numbers mean and how specific cells are defined (AM, DC1, DC2, CD1c-, DC5, pDC).

3. Figure 1 and others – A Table defining the various cell types should be added to provide clarity for the reader. It is very difficult in the various figures to keep track of the definitions of cells based on surface markers shown in the legends of the figures.

4. Figure 1 and others – A problem with the flow cytometry methodology is that it appears that the histograms shown in the various figures are derived from a single analysis in one of the subjects. The Figure Legend to panel 1B says that the plots are representative of n=9 SS BAL and 10 LPS BAL studies. How did the authors cope with variability in the flow cytometry results from subject to subject?

5. Page 8. The authors say that “AM yield in BAL was reduced following LPS inhalation (Figure 1C)”. What is shown in the figure is a reduced percentage. As there was a large increase in recruited cells, they also should also report whether the total number of AM was reduced, which would be a challenge to explain. Presumably, the large increase in newly recruited cells would reduce the percentage of resident AM with little change in their numbers. If the number of AM actually declined, an explanation would be needed.

6. Figure 1E, showing changes in blood leukocytes after saline or LPS inhalation could be moved to the supplementary data, as the number of circulating mononuclear phagocytes did not change with time after the different inhalation challenges.
7. Page 8, referring to Figure 2. The “phenotype” of CD14⁺CD16⁻ MPs is mentioned in the text. The authors should be clear about what they mean by the word “phenotype”, e.g. by surface markers, light microscopy, or some other parameter.
8. Figure 2, panel A. Is the gene expression data controlled for differences in cell counts in the preparations? This should be clarified in the text and methods.
9. Several of the major results are interesting, but not very surprising. For example (page 10), the finding that CD14⁺CD16⁻ MPs in BAL are recruited blood monocytes that have undergone adaptation in the alveolar environment and the finding that CD14⁺CD16⁻ MPs from the LPS challenged lungs expressed LPS response genes and marked expression of chemokine genes. How do these findings advance the understanding of lung inflammatory responses beyond what is already known?
10. Figure 5, page 16. The authors hypothesized that “resident alveolar macrophages orchestrate recruitment of leukocytes into the alveolar spaces following LPS inhalation”. This idea is not new, and the related findings (induction of LPS response genes and production of chemokines) are expected. Other cells (e.g. epithelial cells in the airways) also contribute to leukocyte recruitment, but the study design using the BAL technique only allowed BAL AM to be studied.
11. Page 23. The last sentence of the Discussion seems overstated, as it is not clear what “potential anti-inflammatory molecular targets for therapy” have been identified by this study.
12. Page 23, Methods regarding timing of BAL. The text refers to BAL 8 hr. after LPS inhalation, but the Methods Section mentions BAL 7 hr. after LPS inhalation. This should be clarified.
13. Supplemental materials. The supplemental materials include an Excel file with specific data about differentially expressed genes in Figures 2, 3 and 5, and a Word file with a Figure showing details about the flow cytometry gating and analysis of leukocyte subsets in BAL fluid. The Excel file adds valuable information about the gene expression findings. With respect to the Supplementary Figure, as noted above, the authors should provide either data or comments about the amount of variability from subject to subject that they saw in the BAL leukocyte analyses, particularly in the post-LPS inhalation samples.

Reviewer #1 (Remarks to the Author):

Acute lipopolysaccharide inhalation recruits dendritic cells and monocytes to the alveolar airspace

This is an excellent paper on recruited monocytes and DCs rapidly adopted gene expression profiles following lipopolysaccharide inhalation.

I feel sorry to say that I am a clinician who was involved in numerous LPS inhalation studies but I am not an expert on monocytes and DCs. Taken this into account I can only comment on the clinical data provided in the paper. The lab findings have to be reviewed by an expert on DCs and MPs.

Clinical review:

60 µg LPS is in the dose range of many studies and after 8 hours some clinical and laboratory findings are often recorded. Unfortunately these data are not provided by the authors. It would be helpful if the authors provide data on clinical symptoms after LPS inhalation at least in the supplement.

The following data should be shown:

What side effects (cough and flu like symptoms) did you record?

Symptoms were recorded immediately following inhalation and at 6 hours. We specifically asked about flu-like symptoms, sore throat, cough, wheeze, chest pain, sputum production, nasal secretions and recorded any other symptoms the subjects were experiencing. Symptoms were minor in our cohort compared with previous well-documented studies (Loh LC et al., *Respir Med* 2006). Two LPS and two saline recipients reported symptoms.

Please record data on the increase of body temperature? It is well known that after LPS challenge there is an increase of at least 0.5 celsius after 8 hours.

Body temperature was recorded hourly until 6 hours. Previous studies have typically used a temperature rise of 0.5°C as a clinical indicator of LPS effect (Michel O et al., *J Allergy Clin Immunol* 2001). We measured a mean temperature increase of 0.7°C at 6hrs in subjects inhaling LPS and 0.4°C in subjects inhaling saline.

In addition please provide a figure on leukocyte- and CRP increase after 8 hours and if available after 24 hours to compare the inflammatory response given to previous studies.

We relied principally on blood leukocyte counts as the indicator of LPS effect. As with Loh et al., who used 50µg inhaled LPS, we saw an approximate doubling of the white cell count (Loh LC et al., *Respir Med*, 2006). As with Michel et al., who also used 50µg inhaled LPS, we saw an approximate tripling of neutrophil count (Michel O et al., *J Allergy Clin Immunol*, 2001). The leukocyte count at 24 hours is confounded by the known inflammatory effect of bronchoalveolar lavage (BAL) itself, manifest in our hands as a rise in the control group. However, the leukocyte count is similar again in both groups at 24 hours and, in the LPS

group, the 24-hour leukocyte count is lower than it was at 6 hours. We did not measure CRP.

Taken together, we believe these data indicate that the LPS dose used was efficacious in driving a typical inflammatory response (and that BAL produced a small inflammatory response of its own, as expected). We have added this information to the Methods section (p18) and produced a Supplementary Figure to demonstrate this effect (Supplementary Figure 2). We apologize for omitting this from the original submission.

Since this is a clinical studies with human volunteers the study has to be registered for instance by clinicaltrial.gov. Unfortunately no clinical trial number is provided-.why?

The reviewer raises an important point. Our group has led 4 clinical trials to date, and we have been fully committed to their registration with sites such as clinicaltrials.gov (for our 4 trials the registrations and PMIDs were as follows: ISRCTN 42695423, PMID 23627345; ClinicalTrials.gov NCT01653665, ISRCTN 95325384, PMID 30064991; ClinicalTrials.gov NCT 02085018, ISRCTN07139948, EudraCT 2013-003301-26, manuscript in press; ClinicalTrials.gov NCT01972425, ISRCTN 65937227, manuscript under review). Before this study, and our previous LPS studies where there is no intervention given with therapeutic intent, we have contacted the UK regulatory authorities for guidance on whether our studies require registration. We have consistently been informed that because LPS is being used to provoke inflammation (not as a medicinal product), the studies are not clinical trials and therefore do not require registration.

No data on the power calculation, primary and secondary parameters are provided-why? Why did you challenge ten and not 16 patients?

The objective of this specific part of our study was to examine cellular innate immune responses in more detail, early in the course of acute lung inflammation, paying particular attention to dendritic cell populations. As this was an exploratory objective, it was not possible to perform formal power calculations in advance. Our initial protocol planned to expose 8 volunteers to LPS and 8 to saline. This sample size was based on the ability to detect a flow cytometry signal in LPS BAL with 6 participants per group in previous LPS studies (Brittan M. et al., *European Respiratory Journal*, 2012). After studying the first few volunteers, it was clear the cellular signal in LPS BAL was again striking and reproducible. We did not have concerns about the power to detect the cellular immune response, but sought a major amendment to study additional volunteers to allow sufficient material for gene expression and functional studies.

I recommend that the authors show the complete study protocol in the supplement as it is usually done in consort journals.

We did not include the complete study protocol because, as outlined in R2, this was not a clinical trial. We feel that the protocol contains too much extraneous information on study objectives and parameters unrelated to this manuscript to be valuable to the reader. However we appreciate that the methods section we submitted contains insufficient information for readers interested in the conduct of the study, therefore we have added an

abbreviated protocol in “Supplementary Methods”. The full protocol will be made available to interested readers on request, though we would be happy to provide more detail if this is felt necessary.

Reviewer #2 (Remarks to the Author):

1. The investigators have provided data on the effect of LPS versus saline (control) inhalation in healthy subjects done 8 hours after exposure with bronchoscopy and BAL 8 hours. The purpose was to characterize the mononuclear phagocytes recruited to the air spaces after LPS inhalation. Classical monocytes and two DC subsets were expanded by LPS. The DCs appear to be recruited from the circulating blood. The recruited MPs acquired gene expression profiles that are characteristics of alveolar macrophages. The number of chemoattracted MPs to the lung was a small fraction of circulating MPs so it cannot be determined whether the source was spleen, liver, or bone marrow or just the blood. There was considerable care and effort to do these studies.

2. Although these studies provide new information in the human lung beyond just mouse lung studies, the significance of the research would be greater if they were compared to patients who developed acute bacterial pneumonia. The authors in fact do state in their discussion that their objective is to understand the function of recruited MPs and other cells in the setting of inflammation. LPS is a way to simulate early inflammation but why not study a set of patient with early acute pneumonia for comparison?

We agree with the reviewer that a study in pneumonia would be interesting. However, we strongly believe that it was essential to characterise the response in volunteers first (and that pneumonia is an ideal area to extend this work into in future). Our LPS study has the advantages of knowing the time of onset of the inflammatory stimulus, clear definition of what the initial inflammatory stimulus was, a relatively predictable size of response with time, and an absence of comorbidities or confounding effects of treatment.

We anticipated that the cell populations we were particularly interested in (mononuclear phagocytes) would be recruited in relatively small numbers, with neutrophils the predominant cells infiltrating the alveoli, and we were keen to establish the detailed characteristics in the previously healthy lung first of all.

Pneumonia brings additional ethical challenges in this setting, as patients with pneumonia will, by definition, have ventilation-perfusion mismatch and the potential for hypoxemia. The responsible organism(s) is not always identifiable, the phase of the pneumonia is hard to ascertain, and it is extremely difficult to obtain patients for study before they have had treatments that might confound results. For all these reasons, we committed to an in-depth study of LPS versus control. The challenges listed have given us pause for thought around which population to study next, and we have considered settings like post-operative lung inflammation (where the timing of the provoking stimulus is at least known) or post-bone marrow transplantation lung inflammation (where prognosis is poor and where very little is known about the inflammatory infiltrate). Our thoughts on the next steps are continuing,

with particular reference to patient biological confounding factors, as considered above. However, the novel insights in the current manuscript will undoubtedly improve and rationalize future study design in pathological conditions.

3. The authors close the discussion with the theory that this data can identify potential anti-inflammatory targets, but what is the rationale for impeding or blocking any of these pathways? Are they part of necessary host defense in the presence of pathogen induced pneumonia. If the authors are thinking about ARDS, they should be cautious here as these host responses have evolved to probably enhance host defense.

The reviewer is absolutely correct, the host inflammatory responses observed could be beneficial or detrimental, and dissection of the functional consequences for the host requires far greater study. We have removed the final, speculative part of the final sentence, i.e. the words "...and identifies potential anti-inflammatory molecular targets for therapy" and replaced them with "Dissection of the functional consequences for the host will provide the opportunity to understand both beneficial and detrimental effects of such inflammation" (p17).

4. The manuscript does lack a good paragraph on limitations of these studies.

We are grateful for this opportunity and had contemplated including such a paragraph in the original submission. This has been added to the discussion on p16.

5. I suppose it was too difficult to obtain consent for a later BAL, but a follow up BAL at 24-48 hours would have added more time-dependent information.

As the reviewer points out, a second BAL in the same individual would have posed important ethical concerns. Furthermore, as is evident from our own data (see R1 above) and data in the literature, BAL itself induces inflammation, and we would have required to leave a longer window to be sure that this second inflammatory stimulus had fully receded in the lung. Therefore, we opted to take "baseline" information from the control group, and not to seek ethical approval for a further "recovery" BAL.

Reviewer #3 (Remarks to the Author):

Comments for Authors

The goal of this study was to better understand the subsets of mononuclear cells recruited to the alveolar spaces of human lungs in response to a defined inflammatory stimulus, bacterial lipopolysaccharides (LPS). The authors used a single stimulus that activates the TLR4 pathway and assessed inflammation in the lungs of humans by bronchoalveolar lavage (BAL) at one time (8 hrs) after inhalation of LPS. Blood and lung cells were analyzed by multiparameter flow cytometry and cell sorting, and gene expression in mononuclear cell subsets was analyzed by mRNA analysis. Inhaled sodium chloride solution was used as a comparison for inhaled LPS. The findings are directly relevant to lung inflammation in humans in response to a defined bacterial stimulus.

1. Overall, this is a descriptive study that characterizes mononuclear cell populations in the

lungs of human volunteers at a single time following the inhalation of a defined stimulus (gram negative LPS, TLR4 agonist). Some of the results are new (e.g. the findings related to DC subpopulations), but other aspects are predictable based on existing literature (e.g. that blood MN migrate into the lungs and modify gene expression and that alveolar macrophages contribute to leukocyte recruitment in response to LPS). The authors need to be more clear about how this study advances the understanding of lung inflammatory responses in humans. In addition, do the lung responses appear to differ from what is known in other tissues?

Recruitment of dendritic cells from the blood to the gas exchanging regions of the previously healthy human lung during early inflammation has never been documented before. Most data to date have been generated in rodents which has focused on monocyte recruitment, yet abundant evidence has pointed to striking difference in rodent and human immune responses in recent years (Seok J et al., *PNAS*, 2013). DCs have been shown to accumulate rapidly in the bronchial (conducting) mucosa in atopic individuals following allergen challenge (Jahnsen FL et al., *Thorax*, 2001). In this earlier study, detailed analyses of the infiltrating mononuclear phagocyte populations were not possible. The novelty of our study therefore lies primarily in the capacity of the LPS model to provide relevant information in humans for the first time (rather than extrapolating assumptions from other species).

In our study, we ascertained that more than 1 subset of DCs is recruited. We were able to prove that DCs were functional, we determined that they expressed the surface antigen profile of blood DCs and we were able to examine their transcriptional activity. Gene expression analyses supported our assertion that DCs were derived from blood DCs rather than blood monocytes. This is an important point as the most robust study of human inflammatory DCs to date concluded that they were monocyte-derived (Segura E et al., *Immunity*, 2013). Segura et al studied chronic inflammatory exudates in patients where the onset of inflammatory stimulus is not known. In contrast, our study results suggest direct recruitment of blood DCs during acute lung inflammation.

We are clear that LPS inhalation is a tool to study the earliest events following a defined onset of acute inflammation and not a model of a particular respiratory disease (see revised limitations paragraph). The recruitment of blood DCs to the alveolar airspace during acute inflammation is immunologically significant. DCs recruited in the presence of PAMPs such as LPS are likely to induce a functional rather than tolerogenic response to the peptides they acquire. This could potentially be pathogenic, for example in promoting bronchial atopic diseases associated with household or occupational LPS exposure. This could also potentially be manipulated for therapeutic benefit, for example, using low-dose LPS to generate functional immune responses to inhaled vaccines or to promote tissue accumulation of DC vaccines (though, as reviewer 2 points out, this requires much more dissection before firm clinically applicable conclusions can be drawn). We have amended our discussion section on p14, p15 and p16 to emphasise these points.

2. Table 1. A footnote should be added to the Table to define what the numbers mean and how specific cells are defined (AM, DC1, DC2, CD1c-, DC5, pDC).

A column of definitions and a footnote has been added to Table 1.

3. Figure 1 and others – A Table defining the various cell types should be added to provide clarity for the reader. It is very difficult in the various figures to keep track of the definitions of cells based on surface markers shown in the legends of the figures.

Mononuclear phagocyte nomenclature is difficult to follow and we agree improvements were needed to help the reader keep track. We have added alternative nomenclature and defining antigens into Table 1 and colour-coded the columns to match the relevant flow cytometry gates in Figure 1.

4. Figure 1 and others – A problem with the flow cytometry methodology is that it appears that the histograms shown in the various figures are derived from a single analysis in one of the subjects. The Figure Legend to panel 1B says that the plots are representative of n=9 SS BAL and 10 LPS BAL studies. How did the authors cope with variability in the flow cytometry results from subject to subject?

Figure 1 shows dot plots from single samples of blood, saline BAL and LPS BAL in order to demonstrate how populations are defined and to compare the cellular profiles visually (1B). Summary data from the whole study population is shown in Figure 1C. In other instances where representative flow cytometry plots are shown (Figure 3C, 4B, 4D, 4E), summary data is shown alongside. We found the cellular profiles of BAL consistent, as the error bars in Figure 1C illustrate.

When performing flow cytometry analyses such as these over a number of weeks, it is inevitable that there will be inter-individual variability in the data. We performed careful instrument maintenance (for example running cytometry set up and tracking beads at least daily to ensure consistent performance of all lasers). We adjusted antibody concentrations to the cell volume. We intermittently ran PBMC as a biological control on the analysis template to ensure that gates were capturing the expected populations. We avoided performing analyses that are vulnerable to fluctuations in instrument sensitivity (for example calculating mean fluorescence intensities). We have added this information to the Methods section p19.

The only figures where representative flow cytometry data is shown without accompanying summary data are Figure 3A and Supplementary Figure 1. In both of these, plots are aimed at showing relative antigen expression. Antigen expression was reproducible between subjects, but overlaying histograms from numerous individuals would impair the clarity of the plot. The point we are trying to emphasize Figure 3A, for example is that antigen expression differs between condition (blood, saline BAL, LPS BAL) by several logs.

5. Page 8. The authors say that “AM yield in BAL was reduced following LPS inhalation (Figure 1C)”. What is shown in the figure is a reduced percentage. As there was a large increase in recruited cells, they also should also report whether the total number of AM was reduced, which would be a challenge to explain. Presumably, the large increase in newly recruited cells would reduce the percentage of resident AM with little change in their numbers. If the number of AM actually declined, an explanation would be needed.

Consistent with previous literature, we found that alveolar macrophage yield as a proportion of cells was reduced following LPS inhalation (Michel O, *BMC Pharmacol Toxicol*, 2014; Hernandez ML, *J Allergy Clin Immunol*, 2015). In Figure 1C, we represented the data as absolute cell concentration (number per volume BAL). In the previous studies mentioned, absolute cell numbers remained static following LPS inhalation so it was important to point out our result appeared different. We rigorously cross-checked analysis and found a formula error (proportions were being applied to one sample leukocyte count rather than the leukocyte count of each sample). This error has been corrected with the figure and the text adjusted accordingly. We thank the reviewer for helping us identify this issue. Absolute numbers of alveolar macrophages are not significantly different following LPS inhalation. We also took the opportunity to reorder the plots in Figure 1C in order of descending frequency in BAL, to make interpretation easier.

6. Figure 1E, showing changes in blood leukocytes after saline or LPS inhalation could be moved to the supplementary data, as the number of circulating mononuclear phagocytes did not change with time after the different inhalation challenges.

We respectfully request that these data be left in Figure 1E. As per the comments of reviewer 1, we feel it is useful to see the neutrophil count to help compare consistency with other LPS studies. The data also demonstrate the differing dynamics of recruited populations following LPS.

7. Page 8, referring to Figure 2. The “phenotype” of CD14⁺⁺CD16⁻ MPs is mentioned in the text. The authors should be clear about what they mean by the word “phenotype”, e.g. by surface markers, light microscopy, or some other parameter.

We agree that “phenotype” could be interpreted to mean several things. We have amended the text on p6 and p8 to be clear that by phenotype we mean antigen expression as measured by flow cytometry.

8. Figure 2, panel A. Is the gene expression data controlled for differences in cell counts in the preparations? This should be clarified in the text and methods.

Gene expression analysis was performed on lysates containing consistent numbers of cells ($1-2 \times 10^4/5\mu\text{l}$). During analysis, the nSolver normalisation pipeline utilised mRNA counts from synthetic spike-in controls and a combination of 15 housekeeping genes to give a sample-specific correction factor to all probes (genes), reducing technical variability between samples, for example pipetting errors and sample input variability. We agree this wasn't clearly stated in our Methods section and we are grateful for the opportunity to clarify this (p26).

9. Several of the major results are interesting, but not very surprising. For example (page 10), the finding that CD14⁺⁺CD16⁻ MPs in BAL are recruited blood monocytes that have undergone adaptation in the alveolar environment and the finding that CD14⁺⁺CD16⁻ MPs from the LPS challenged lungs expressed LPS response genes and marked expression of

chemokine genes. How do these findings advance the understanding of lung inflammatory responses beyond what is already known?

It is logical that a CD14⁺⁺CD16⁻ tissue infiltrate will originate from blood classical monocytes but few studies in humans have actually attempted to demonstrate this. In most instances tissue inflammatory composition has been examined in isolation without reference to blood and origins have been assumed (Lowes MA. et al., *PNAS*, 2005; Jahnsen FL. et al., *Thorax* 2001; Brittan M. et al., *European Respiratory Journal* 2014).

Due to our referencing of blood it seems obvious that BAL CD14⁺⁺CD16⁻ MPs will be monocytes recruited from blood with tissue adaptation, but the nature of tissue adaptation required investigation. In both mouse and human studies, monocytes recruited to tissues can acquire DC or macrophage characteristics (Leon B et al., *Immunity*, 2007; Bain CC. et al., *Mucosal Immunology*, 2013; Segura E et al., *Immunity*, 2013). Monocytes have also been described entering tissue with minimal adaptation (Jakubzick C. et al., *Immunity* 2013). The type of MP characteristics adopted is highly relevant to the subsequent immune response as macrophages and DCs differ in capacity for migration, phagocytosis, ability to induce adaptive immune responses and in the cytokine/chemokine secretion profile. In both steady-state and acute inflammation, the MPs we studied in LPS BAL had adopted macrophage characteristics, for example the expression of phagocytic receptor genes *MRC1*, *FCGR3A/B* and *MSR1*.

We agree it is not surprising that MPs derived from classical monocytes are responsive to LPS. To us, the key point from examining the gene expression profile of CD14⁺⁺CD16⁻ BAL MPs recruited in inflammation from those present in steady state was that the same cell type functions to maintain quiescence during one condition but operates to amplify monocyte, T cell, DC and neutrophil recruitment in another.

10. Figure 5, page 16. The authors hypothesized that “resident alveolar macrophages orchestrate recruitment of leukocytes into the alveolar spaces following LPS inhalation”. This idea is not new, and the related findings (induction of LPS response genes and production of chemokines) are expected. Other cells (e.g. epithelial cells in the airways) also contribute to leukocyte recruitment, but the study design using the BAL technique only allowed BAL AM to be studied.

There is a body of literature demonstrating that alveolar macrophages are critically important in maintaining alveolar immune quiescence, thus preventing the fragile gas exchange surface from the damaging products of inflammation (Holt PG., *Am Rev Respir Dis* 1978; Thepen T., *J Exp Med* 1978; Elder A., *Exp Lung Res*, 2005). Depletion of alveolar macrophages increases inflammatory cell recruitment upon LPS exposure (Elder A., *Exp Lung Res*, 2005). However, alveolar macrophages can orchestrate inflammation and their activity has been implicated in lung pathology (Huang X, *Mediators Inflamm.*, 2018). Macrophage activity is now understood to exist in a spectrum of states rather simply flipping between inflammatory (M1) and anti-inflammatory (M2) states (Xue J., *Immunity* 2014), so we thought it important to characterize the consequences of in vivo LPS exposure on macrophage transcriptional activity. To our knowledge, no other studies have achieved this in humans to date.

We accept that the alveolar macrophage is only one component of the interface between air and alveolus. We have made this caveat clearer in our revised paragraph on limitations. In vitro studies have suggested that both types of alveolar epithelial cells are responsive to LPS (Wong MH, *PLoS One*, 2013). However, it is not possible to isolate alveolar epithelial cells and study them with the techniques we have used to understand alveolar macrophages. Isolation from healthy volunteers would involve endobronchial biopsy, which most investigators believe involves too much clinical risk to be ethical in healthy volunteers. Furthermore, the alveolar epithelial cell yield from such a biopsy would be inadequate for NanoString gene expression analysis.

11. Page 23. The last sentence of the Discussion seems overstated, as it is not clear what “potential anti-inflammatory molecular targets for therapy” have been identified by this study.

Thank you for pointing this out, in keeping with reviewer 2 (C6) and we have removed this sentence, as described in R6.

12. Page 23, Methods regarding timing of BAL. The text refers to BAL 8 hr. after LPS inhalation, but the Methods Section mentions BAL 7 hr. after LPS inhalation. This should be clarified.

BAL was performed at 8 hours. The erroneous mention of 7 hours (for which we apologize) has been removed from the text.

13. Supplemental materials. The supplemental materials include an Excel file with specific data about differentially expressed genes in Figures 2, 3 and 5, and a Word file with a Figure showing details about the flow cytometry gating and analysis of leukocyte subsets in BAL fluid. The Excel file adds valuable information about the gene expression findings. With respect to the Supplementary Figure, as noted above, the authors should provide either data or comments about the amount of variability from subject to subject that they saw in the BAL leukocyte analyses, particularly in the post-LPS inhalation samples.

This comment is answered above in response to Reviewer 3 comment 4.

REVIEWERS' COMMENTS:

Reviewer #1 (Remarks to the Author):

This is an excellent paper which targets an important field of regulation of neutrophilic inflammation via dendritic cells in lung disease.

The revised version is significantly clearer for the critical readers and now ready for publication.

Reviewer #2 (Remarks to the Author):

The responses to the comments and questions that I raised are clear and thoughtful. The limitations paragraph is now included in the discussion which provides more balance. Although the studies are primarily descriptive, I think they have substantial value to the field of understanding the human (and pulmonary) responses to pathogens.

Michael A. Matthay MD

Reviewer #3 (Remarks to the Author):

The authors have provided detailed responses to the critical comments in the prior review.

Comment 1. The authors have revised parts of the Discussion to focus more on their original data about the dendritic cell subsets recruited to the lungs in response to inhaled LPS, as this is the most novel data in the study. However, at line 547 of the Discussion, they should reconsider their statement "We propose a role for resident AMs in mediating blood classical monocyte, DC subset and neutrophil recruitment into the alveolar space". This claim is too broad. The role of AMs in recruiting MN and MN into the lungs in response to bacterial signals such as LPS has been recognized for a long time. They are proposing to expand this role to include the recruitment of blood DCs – this is the major contribution of their study.

Comment 2. Satisfactory response (modification of Table 1), but I did not find Table 1 in the resubmitted materials to review.

Comment 3. Satisfactory response.

Comment 4. Satisfactory response. Methods section regarding flow cytometry has been improved.

Comment 5. Satisfactory response. Figure 1 has been amended and an error in the data calculations has been corrected.

Comment 6. The authors request that the blood leukocyte data remain in the figure. As the authors note, the PMN data do allow a comparison with other studies, so this is a satisfactory response.

Comment 7. Satisfactory response.

Comment 8. Satisfactory response.

Comment 9. Satisfactory response in the rebuttal, but it would be useful to briefly add the points in paragraphs 1 and 2 of their rebuttal to the Discussion.

Comment 10. The authors have revised the text in response to this comment. The point remains

that the role of AM in recruiting circulating PMN and MN into the lungs in response to pro-inflammatory stimuli such as LPS is already known from prior literature. This study contributes additional information by showing that AM also recruit DCs into the lungs in a relatively short time frame (within 8 hr. of exposure to the stimulus). An additional valuable point is that this entire study was done in humans, so the results are immediately relevant to understanding human biology. Responding to comment 1 above will address the essence of this point.

Comment 11. Satisfactory response. The conclusion of the Discussion has been modified.

Comment 12. Satisfactory

Comment 13. Satisfactory

Additional comments

1. Line 32. "Little is known about the MP repertoire in acute pulmonary inflammation". This statement is not true and should be revised, because it is too broad. We need to know more about DCs and their recruitment to the lungs during pulmonary inflammatory responses. This point is the focus of this paper.

2. Line 43. "Our study defines the kinetics of human DC and monocyte recruitment into the lungs". This is not true, as only one time was studied after LPS inhalation (8 hr.), so a statement about kinetics cannot be made. The study defines the characteristics of recruited DCs and MN, which is valuable.

3. Lines 106-108. This sentence is confusing and should be edited for clarity.

4. Line 190. The text mentions the goal of the blood studies was to "define the dynamics of monocyte and DC recruitment to the alveolar space", but the data do not do this. Making measurements at several times in blood, which showed no changes aside from an increase in PMN, do not provide insight about cellular recruitment into the lungs. This statement should be revised. Serial BAL measurements would be required to study the dynamics of cellular recruitment into the alveolar spaces, which is clearly not feasible in the context of this human study.

5. Line 220. Typo "principle" should be "principal"

6. Figure 1. This Figure has been revised and has been improved. Panel C y axis now shows absolute cell counts, showing the large increase in PMN without a significant increase in AM. A remaining question is why the number of pDCs is not significantly increased (Panels B and C) – could this be a false negative because of the relatively small number of samples?

7. Figure 3. Panel E. Minor points. Explain the color codes at the top of the columns. Presumably the colors correspond to the colors in Panel B, but this should be explicitly stated in the legend. In addition, explain why there are 11 orange coded columns (for LPS/BAL) when only 10 subjects were studied.

8. Figure 4. A larger version of Panel 4Ai should be made available in the Supplementary Materials. The labels are very small in the printed text. Panel 4Dii – explain what "14" means on the x axis of the bar graphs. Panel 4D should be called out in the text, perhaps in line 339. In Panel 4E, the boxes denoting the regions of interest should be explained in the legend.

9. Figure 5. The legend to panel B contains a typo ("LS", should be "LPS" BAL AM).

10. Discussion. Line 468. The statement that the authors observed recruitment of around 2 million monocytes is hard to understand. The LPS was inhaled into the entire lungs, but the BAL

procedure sampled only one relatively small area. The calculations behind this statement should be explained, or the statement should be deleted.

11. Supplementary Table 1. The spirometry data are given as volumes and the volumes are smaller in the saline group. The % predicted values should be included because there were more women in the saline group, and they would be expected to have smaller lung volumes.

REVIEWERS' COMMENTS:

Reviewer #1 (Remarks to the Author):

This is an excellent paper which targets an important field of regulation of neutrophilic inflammation via dendritic cells in lung disease.

The revised version is significantly clearer for the critical readers and now ready for publication.

Reviewer #2 (Remarks to the Author):

The responses to the comments and questions that I raised are clear and thoughtful. The limitations paragraph is now included in the discussion which provides more balance. Although the studies are primarily descriptive, I think they have substantial value to the field of understanding the human (and pulmonary) responses to pathogens.

Michael A. Matthay MD

Reviewer #3 (Remarks to the Author):

The authors have provided detailed responses to the critical comments in the prior review.

Comment 1. The authors have revised parts of the Discussion to focus more on their original data about the dendritic cell subsets recruited to the lungs in response to inhaled LPS, as this is the most novel data in the study. However, at line 547 of the Discussion, they should reconsider their statement "We propose a role for resident AMs in mediating blood classical monocyte, DC subset and neutrophil recruitment into the alveolar space". This claim is too broad. The role of AMs in recruiting MN and MN into the lungs in response to bacterial signals such as LPS has been recognized for a long time. They are proposing to expand this role to include the recruitment of blood DCs – this is the major contribution of their study.

The sentence in question has been revised to clarify that the novel data pertains to DC recruitment "We propose that resident AMs operate to recruit DCs into the alveolar airspace, in addition to their role in recruiting neutrophils and monocytes." (page 16).

Comment 2. Satisfactory response (modification of Table 1), but I did not find Table 1 in the resubmitted materials to review.

I apologize for omitting Table 1 from the resubmitted materials. I believe that it was distributed shortly afterwards and that the reviewers have had the opportunity to consider the changes.

Comment 3. Satisfactory response.

Comment 4. Satisfactory response. Methods section regarding flow cytometry has been improved.

Comment 5. Satisfactory response. Figure 1 has been amended and an error in the data calculations has been corrected.

Comment 6. The authors request that the blood leukocyte data remain in the figure. As the authors note, the PMN data do allow a comparison with other studies, so this is a satisfactory response.

Comment 7. Satisfactory response.

Comment 8. Satisfactory response.

Comment 9. Satisfactory response in the rebuttal, but it would be useful to briefly add the points in paragraphs 1 and 2 of their rebuttal to the Discussion.

We agree that the points were made more clearly in the rebuttal than they were in the Discussion, so the text has been revised on page 13 to address this.

Comment 10. The authors have revised the text in response to this comment. The point remains that the role of AM in recruiting circulating PMN and MN into the lungs in response to pro-inflammatory stimuli such as LPS is already known from prior literature. This study contributes additional information by showing that AM also recruit DCs into the lungs in a relatively short time frame (within 8 hr. of exposure to the stimulus). An additional valuable point is that this entire study was done in humans, so the results are immediately relevant to understanding human biology. Responding to comment 1 above will address the essence of this point.

I hope that our revision to page 16 (as suggested in comment 1) will convey to the reader that DC recruitment is the novel finding of this study.

Comment 11. Satisfactory response. The conclusion of the Discussion has been modified.

Comment 12. Satisfactory

Comment 13. Satisfactory

Additional comments

1. Line 32. "Little is known about the MP repertoire in acute pulmonary inflammation". This statement is not true and should be revised, because it is too broad. We need to know more about DCs and their recruitment to the lungs during pulmonary inflammatory responses. This point is the focus of this paper.

This sentence has been amended to "little is known about DC recruitment in acute pulmonary inflammation" (page 1).

2. Line 43. "Our study defines the kinetics of human DC and monocyte recruitment into the lungs". This is not true, as only one time was studied after LPS inhalation (8 hr.), so a statement about kinetics cannot be made. The study defines the characteristics of recruited DCs and MN, which is valuable.

This sentence has been amended to "Our study defines the characteristics of human DCs and monocytes recruited into bronchoalveolar space immediately following localised acute inflammatory stimulus in vivo." (page 1-2).

3. Lines 106-108. This sentence is confusing and should be edited for clarity.

This sentence has been divided and clarified: "Briefly, this confirmed the presence of cDC1 (DC1) and pDC (pDC or DC6). It revealed two subdivisions within cDC2 (DC2, DC3) and identified additional DCs subsets: Axl⁺Siglec-6⁺ DCs (AS DC or DC5) and CD1c⁻CD141⁻ DCs (DC4)" (page 3). In combination with the surface phenotype information added to Table 1, we anticipate this will help orient the reader to DC subsets.

4. Line 190. The text mentions the goal of the blood studies was to "define the dynamics of monocyte and DC recruitment to the alveolar space", but the data do not do this. Making measurements at several times in blood, which showed no changes aside from an increase in PMN, do not provide insight about cellular recruitment into the lungs. This statement should be revised. Serial BAL measurements would be required to study the dynamics of cellular recruitment into the alveolar spaces, which is clearly not feasible in the context of this human study.

Yes, it is incorrect to refer to this as dynamics of recruitment for the reasons stated by Reviewer 3. The sentence has been amended to "To examine the impact of acute lung inflammation on peripheral blood MP populations, their concentrations were tracked at 2-hour intervals following inhalation of LPS or saline" (page 6).

5. Line 220. Typo "principle" should be "principal"

Thank you for identifying this error. It has been amended.

6. Figure 1. This Figure has been revised and has been improved. Panel C y axis now shows absolute cell counts, showing the large increase in PMN without a significant increase in AM. A remaining question is why the number of pDCs is not significantly increased (Panels B and C) – could this be a false negative because of the relatively small number of samples?

The trend was towards an increase in BAL pDCs following LPS inhalation, but this trend did not reach significance, with $p=0.063$ by unpaired t-test of SS-BAL vs. LPS BAL. It is possible that this magnitude of increase would reach statistical significance with a greater number of samples. Within the constraints of study design, we could identify a difference in the recruitment of DC subsets, with DC2/3 and DC5 robustly recruited and pDC recruitment probably best considered minor rather than negative.

7. Figure 3. Panel E. Minor points. Explain the color codes at the top of the columns.

Presumably the colors correspond to the colors in Panel B, but this should be explicitly stated in the legend. In addition, explain why there are 11 orange coded columns (for LPS/BAL) when only 10 subjects were studied.

The Figure has been edited to clarify the samples in the heatmap. Thank you for identifying that the numbers of columns in the heatmap has not been adequately explained. Good quality expression data from 7 LPS BAL recipients was obtained. Three samples were sorted as total DC2/3 and four were sorted into two fractions, yielding 11 expression datasets from 7 recipients. This detail is now more transparent with the explanation of our sorting strategy in Supplementary Figure 3.

8. Figure 4. A larger version of Panel 4Ai should be made available in the Supplementary Materials. The labels are very small in the printed text. Panel 4Dii – explain what “14” means on the x axis of the bar graphs. Panel 4D should be called out in the text, perhaps in line 339. In Panel 4E, the boxes denoting the regions of interest should be explained in the legend.

A larger version of Panel 4Ai has been made available as Supplementary Figure 4. Explanation of the abbreviation “14” has been added to the figure legend. A link between Figure 4 panel D and the text has been added on page 10. Panel 4E has been explained more fully in the legend.

9. Figure 5. The legend to panel B contains a typo (“LS”, should be “LPS” BAL AM).

Thank you for identifying this error. It has been amended.

10. Discussion. Line 468. The statement that the authors observed recruitment of around 2 million monocytes is hard to understand. The LPS was inhaled into the entire lungs, but the BAL procedure sampled only one relatively small area. The calculations behind this statement should be explained, or the statement should be deleted.

The statements pertaining to numbers of recruited monocytes and DCs have been removed from the discussion. It was incorrect of us to estimate that the number of recruited cell ‘A’ is represented by the number of cell ‘A’ in LPS BAL minus the number of cell ‘A’ in saline BAL. Even though our sampling method was consistent, it did only sample a small area while the stimulus was global.

11. Supplementary Table 1. The spirometry data are given as volumes and the volumes are smaller in the saline group. The % predicted values should be included because there were more women in the saline group, and they would be expected to have smaller lung volumes.

The % predicted values have been added to Supplementary Table 1.